# **AutoGen**: Enabling Next-Gen LLM Applications via Multi-Agent Conversations

**Qingyun Wu**[*]
Penn State University
qingyun@autogen.team

**Gagan Bansal**
Microsoft Research
gaganbansal@microsoft.com

**Jieyu Zhang**
University of Washington
jieyuz2@cs.washington.edu

**Yiran Wu**
Penn State University
yiran.wu@psu.edu

**Beibin Li**
Microsoft Research
beibin.li@microsoft.com

**Erkang Zhu**
Microsoft Research
erkang.zhu@microsoft.com

**Li Jiang**
Microsoft
lijiang1@microsoft.com

**Xiaoyun Zhang**
Microsoft
xiaoyun.zhang@microsoft.com

**Shaokun Zhang**
Penn State University
shaokun.zhang@psu.edu

**Jiale Liu**
Penn State University
jjl7199@psu.edu

**Ahmed Awadallah**
Microsoft Research
hassanam@microsoft.com

**Ryen W. White**
Microsoft Research
ryenw@microsoft.com

**Doug Burger**
Microsoft Research
dburger@microsoft.com

**Chi Wang***
Microsoft Research
chi@autogen.team

## Abstract

We present AutoGen,[1] an open-source framework that allows developers to build LLM applications by composing multiple *agents* to converse with each other to accomplish tasks. AutoGen agents are customizable, *conversable*, and can operate in various modes that employ combinations of LLMs, human inputs, and tools. It also enables developers to create flexible agent behaviors and conversation patterns for different applications using both natural language and code. AutoGen serves as a generic infrastructure and is widely used by AI practitioners and researchers to build diverse applications of various complexities and LLM capacities. We demonstrate the framework's effectiveness with several pilot applications, on domains ranging from mathematics and coding to question-answering, supply-chain optimization, online decision-making, and entertainment.

## 1  Introduction

Large language models (LLMs) are becoming a crucial building block in developing powerful *agents* that utilize LLMs for reasoning, tool usage, and adapting to new observations (Yao et al., 2022; Xi et al., 2023; Wang et al., 2023). As the scope and complexity of tasks suitable for LLMs increase, a natural strategy for enhancing agent capabilities is to employ multiple cooperating agents. Prior work suggests that multiple agents can help encourage divergent thinking (Liang et al., 2023), improve factuality and reasoning (Du et al., 2023; Naik et al., 2023), and provide guardrails (Wu et al., 2023). Given the early promising evidence, an intriguing question is: How can we facilitate the development of LLM applications that span a broad spectrum of domains and complexities using a multi-agent approach? Our insight

---

[*]Equal contribution
[1]https://github.com/microsoft/autogen

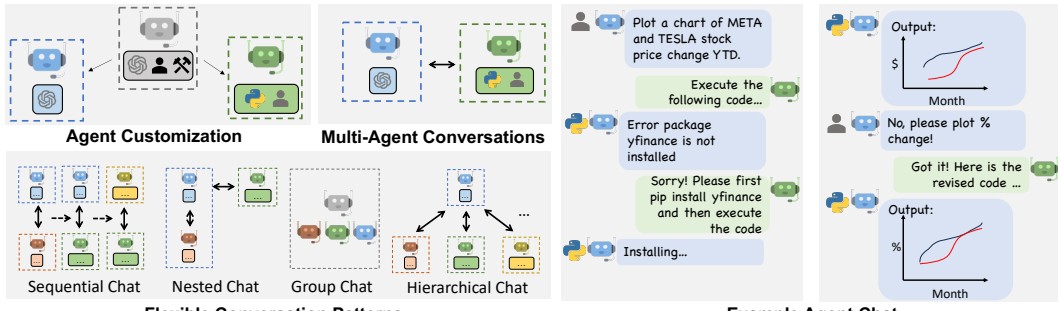

Figure 1: `AutoGen` enables diverse LLM-based applications using multi-agent conversations. (Top-left) `AutoGen` agents are conversable, customizable, and can be based on LLMs, tools, humans, or even a combination of them. (Top-middle) Agents can converse to solve tasks. (Right) They can form a chat, potentially with humans in the loop. (Bottom-left) The framework supports flexible conversation patterns.

is to use multi-agent *conversations*. There are at least three reasons confirming its general feasibility and utility, thanks to recent advances in LLMs: First, chat-optimized LLMs, such as GPT-4, demonstrate ability to incorporate feedback. LLM agents can cooperate through conversations with each other or humans, for example, in a dialogue where agents provide and seek reasoning, observations, critiques, and validation. Second, because a single LLM can exhibit a broad range of capabilities, conversations between differently configured agents can help combine these broad LLM capabilities in a modular and complementary manner. Third, LLMs have demonstrated the ability to solve complex tasks when broken into simpler sub-tasks. Multi-agent conversations can intuitively facilitate this partitioning and integration.

We desire a multi-agent conversation framework with generic abstraction and effective implementation that has the flexibility to satisfy different application needs. Achieving this requires addressing two critical questions: (1) How to design individual agents that are capable, reusable, customizable, and effective in multi-agent collaboration? (2) How can we develop a straightforward, unified interface that accommodates a wide range of agent conversation patterns? In practice, applications of varying complexities may need distinct sets of agents with specific capabilities and may require different conversation patterns, such as single- or multi-turn dialogues, different human involvement modes, and static vs. dynamic conversations. Moreover, developers may prefer the flexibility to program agent interactions in natural language or code. We present `AutoGen`, a generalized multi-agent conversation framework (Figure 1), based on the following new concepts:

1  **Customizable and conversable agents.** `AutoGen` uses a generic agent design that can leverage LLMs, human inputs, tools, or a combination thereof. Developers can conveniently create agents with different roles or responsibilities by selecting and configuring a subset of built-in capabilities or defining new capabilities. To make these agents suitable for multi-agent conversation, every agent is made *conversable* – they can receive, react, and respond to messages. When configured properly, an agent can hold multiple turns of conversations with other agents autonomously or with humans in the loop. The conversable agent design leverages the strong capability of the most advanced LLMs in taking feedback and making progress via conversation, and also allows combining capabilities of LLMs in a modular fashion. (Section 2.1)

2  **Conversation programming.** A fundamental insight of `AutoGen` is to simplify and unify complex LLM applications as multi-agent conversations. Thus, `AutoGen` adopts a programming paradigm centered around these inter-agent conversations. We refer to this paradigm as *conversation programming*, which streamlines the development of intricate applications via two primary steps: (1) defining a set of conversable agents with specific capabilities and roles; (2) programming the interaction behavior between agents via conversation-centric *computation* and *control*. Both steps can be achieved via a fusion of

natural and programming languages. `AutoGen` provides ready-to-use implementations and also allows easy extension and experimentation for both steps. (Section 2.2)

We offer a suite of multi-agent applications realized with `AutoGen`, showcasing the framework's ability to support applications of varied complexities. With these applications, we demonstrate `AutoGen`'s potential to significantly enhance task completion performance and innovate LLM usage while minimizing development effort. Beyond the demonstrated applications, `AutoGen` has also seen widespread adoption in the wild, fostering a vibrant and active community.

**Related Work.** Several contemporaneous explorations of multi-agent approaches exist, including Generative Agents (Park et al., 2023), multi-agent debate (Liang et al., 2023; Du et al., 2023), CAMEL (Li et al., 2023b), BabyAGI (BabyAGI, 2023), MetaGPT (Hong et al., 2023), ChatDev (Qian et al., 2023), AgentVerse (Chen et al., 2023b), AutoAgents (Chen et al., 2023a). These systems are designed for specific types of scenarios or problem-solving paradigms, which limits their flexibility and generalizability as comprehensive frameworks. For instance, MetaGPT and ChatDev prioritize software engineering tasks and only support certain multi-agent structures, such as chains or Standardized Operating Procedures. AgentVerse primarily simulates the problem-solving processes of a human group following a sequence of pre-defined stages. CAMEL supports multi-agent systems with two or three agents following a fixed workflow pattern. One notable difference of AutoGen is that it supports diverse workflows because of its composable conversation patterns and does not explicitly restrict the number of agents. We include an expanded discussion of this related work and single-agent systems/frameworks in Appendix B.

## 2 The `AutoGen` Framework

To reduce the effort required for developers to create complex LLM applications across various domains, a core design principle of `AutoGen` is to streamline them using multi-agent conversations. This approach also aims to maximize the reusability of implemented agents. This section introduces the two key concepts of `AutoGen`: conversable agents and conversation programming.

### 2.1 Conversable Agents

In `AutoGen`, a *conversable agent* is an entity with a specific role that can send message to and receive message from the other conversable agents, e.g., to start or continue a conversation. It maintains its internal context based on sent and received messages and can be configured to possess a set of capabilities, e.g., enabled by LLMs, tools, human input, etc. The agents can act according to the programmed behavior patterns described next.

**Agent capabilities powered by LLMs, humans, and tools.** `AutoGen` allows flexibility to equip its agents with various capabilities, which directly affect how it processes and responds to messages. The built-in composable agent capabilities include: **1) LLMs.** LLM-backed agents utilize advanced capabilities such as role-playing, implicit state inference, making progress based on conversation history, and coding. These capabilities can be combined and enhanced in different ways via novel prompting techniques[2]. `AutoGen` also offers enhanced LLM inference features such as result caching, error handling, message templating, etc., via an enhanced LLM inference layer. **2) Humans.** Human involvement is desired or even essential in many LLM applications. `AutoGen` lets a human participate in agent conversation via human-backed agents, which could solicit human inputs at certain rounds of a conversation depending on the agent configuration. The default *user proxy* agent allows *configurable* human involvement levels and patterns, e.g., frequency and conditions for requesting human input including the option for humans to skip providing input. **3) Tools.** Tool-backed agents have the capability to execute tools via code execution or function

---

[2]Appendix D presents an example of such novel prompting techniques which empowers the default LLM-backed assistant agent in `AutoGen` to converse with other agents in multi-step problem-solving.

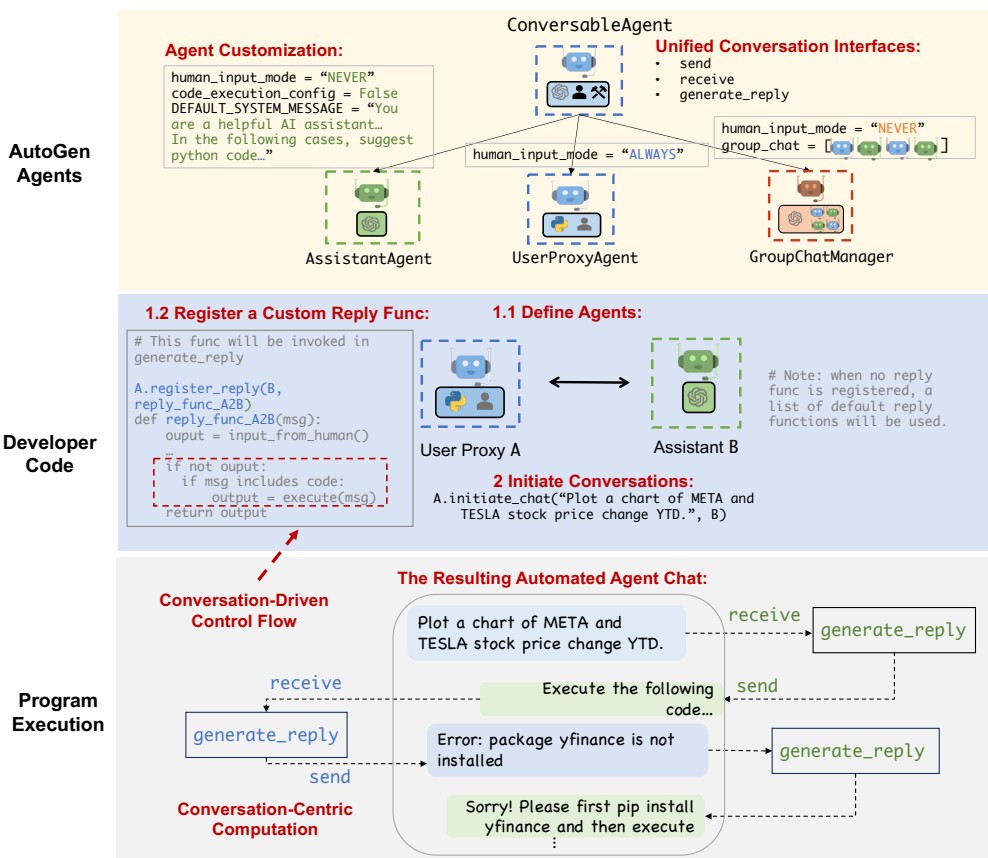

Figure 2: Illustration of how to use AutoGen to program a multi-agent conversation. The top sub-figure illustrates the built-in conversable agents provided by AutoGen. The middle sub-figure shows an example of using AutoGen to develop a two-agent system with a custom reply function. The bottom sub-figure illustrates the resulting automated agent chat from the two-agent system during program execution.

execution. For example, the default user proxy agent in AutoGen is able to execute code suggested by LLMs, or make LLM-suggested function calls.

**Agent customization.** Based on application-specific needs, each agent can be configured to have a mix of basic back-end types to exhibit complex behavior in multi-agent conversations. AutoGen allows easy creation of agents with specialized capabilities and roles by reusing or extending the built-in agents. The yellow-shaded area of Figure 2 provides a sketch of the built-in agents in AutoGen. The ConversableAgent class is the most basic agent abstraction and, by default, can use LLMs, humans, and tools. The AssistantAgent and UserProxyAgent are two pre-configured ConversableAgent subclasses, each representing a common usage mode, i.e., acting as an AI assistant (backed by LLMs) and acting as a human proxy to solicit human input or execute code/function calls (backed by humans and/or tools). In the example on the right-hand side of Figure 1, an LLM-backed assistant agent and a tool- and human-backed user proxy agent are deployed together to tackle a task. Here, the assistant agent generates a solution with the help of LLMs and passes the solution to the user proxy agent. Then, the user proxy agent solicits human inputs or executes the assistant's code and passes the results as feedback back to the assistant. One can compose a complex agent using nested chat (introduced in the next subsection) among simpler agents and increae the complexity recursively.

## 2.2 Conversation Programming

To develop applications where agents make meaningful progress on tasks, developers also need to be able to specify and properly control these multi-agent conversations. To this end, AutoGen utilizes *conversation programming*, a paradigm that concerns two concepts: the first is *computation* – the actions agents take to compute their response in a multi-agent conversation. And the second is *control flow* – the order and conditions under which individual computations in the conversation are executed or evaluated. As we will show in the applications section, the ability to program these helps implement many flexible multi-agent conversation patterns. In AutoGen, agent computations are conversation-centric. An agent takes actions based on conversations it is involved in, and these actions further lead to message passing for subsequent conversations. Similarly, control flow is conversation-driven – the participating agents' decisions on which agents to send messages to and the procedure of computation are functions of the inter-agent conversation. This paradigm facilitates intuitive reasoning about complex workflows through actions of agents and message-passing between agents.

Figure 2 provides a simple illustration. The middle sub-figure shows how each individual agent performs its role-specific, conversation-centric computations to generate responses (e.g., via LLM inference calls and code execution).The bottom sub-figure demonstrates a conversation-based control flow. When the assistant receives a message, the user proxy agent generates a reply based on code execution or solicits human inputs. The task progresses through conversations displayed in the dialog box. AutoGen features the following design patterns to facilitate conversation programming.

**Unified interfaces and auto-reply mechanisms for automated agent chat.** Agents in AutoGen have unified conversation interfaces for performing the corresponding conversation-centric computation. Those low-level interfaces include:

- `send`/`receive` for sending/receiving messages; and
- `generate_reply` for taking actions and generating a response based on the received message;
- `register_reply` for registering custom reply function.

AutoGen also introduces and by default adopts an **agent auto-reply** mechanism to realize conversation-driven control: Once an agent receives a message from another agent, it automatically invokes `generate_reply` and sends the reply back to the sender unless a termination condition is satisfied. AutoGen provides built-in reply functions based on LLM inference, code or function execution, or human input. One can also register custom reply functions (via the `register_reply` interface) to customize the behavior pattern of an agent, e.g., to chat with another agent before replying to the sender agent realizing the *nested chat* conversation pattern. Under this mechanism, once the reply functions are registered, and the conversation is initialized, the conversation flow is naturally induced, and thus the agent conversation proceeds naturally without any extra control plane, i.e., a special module that controls the conversation flow. For example, with the developer code in the blue-shaded area (marked "Developer Code") of Figure 2, one can readily trigger the conversation among the agents, and the conversation would proceed automatically, as shown in the dialog box in the grey shaded area (marked "Program Execution") of Figure 2. The auto-reply mechanism provides a decentralized, modular, and unified way to define the workflow.

**Control by fusion of programming and natural language.** AutoGen allows the usage of programming and natural language in various control flow management patterns:

- Natural-language control via LLMs: One can control the conversation flow by prompting LLM-backed agents with natural language. For instance, the default system message of the built-in AssistantAgent uses natural language to instruct agents to write code and debug when needed. It also guides the agent to confine LLM outputs, making it easier for other agents to consume. More examples of such controls can be found in Appendix D.
- Programming-language control: In AutoGen, Python code can be used to specify the termination condition, human input mode, and tool execution logic, e.g., the max number

of auto replies. One can also register programmed auto-reply functions to control the conversation flow with Python code, as shown in the code block identified as "Conversation-Driven Control Flow" in Figure 2.

- Control transition between natural and programming language: `AutoGen` also supports flexible control transition between natural and programming language. One can achieve transition from code to natural-language control by invoking an LLM inference containing certain control logic in a customized reply function; or transition from natural language to code control via LLM-proposed function calls (Eleti et al., 2023).

**Composable conversation patterns.** The conversation programming paradigm enables the composition of multi-agent conversations with diverse patterns, both statically and dynamically. For enhanced usability, we provide interfaces for constructing several commonly used conversation patterns, including *two-agent chat, sequential chats, nested chat* and *group chat*. We provide the detailed interfaces for specifying these patterns in Appendix D. Beyond these built-in patterns, one can employ these higher-level interfaces – and the low-level interfaces such as `register_reply` if necessary – recursively to compose more complex and creative patterns, e.g., a nested chat with a group chat nested within, allowing one agent to create its inner monologue, realizing the Society of Mind idea from Minsky (1988). The composed conversation workflow can be static or dynamic. `AutoGen` provides a few general ways to achieve dynamic conversation flows: 1) custom reply functions and triggers. Nested chat and group chat are examples of conversation patterns using built-in custom reply functions. In nested chat, one agent can hold the current conversation while invoking conversations with other agents depending on the content of the current message and context. In group chat, one can define the speaker transition conditions based on the current conversation status. 2) LLM-driven function calls, in which a language model decides whether or not to call a particular function depending on the conversation status.

## 3 Applications of `AutoGen`

We demonstrate six applications using `AutoGen` (see Figure 3). These applications are selected based on their real-world relevance (A1, A2, A4, A5, A6), problem difficulty (A1, A2, A3, A4), and innovative potential (A5, A6). Together, these criteria showcase `AutoGen`'s role in advancing the LLM application landscape. All the applications and their evaluations presented in this section are based on code version 0.1.1 of the library, with code and instructions provided in the Reproducibility Statement section at the end of this paper.

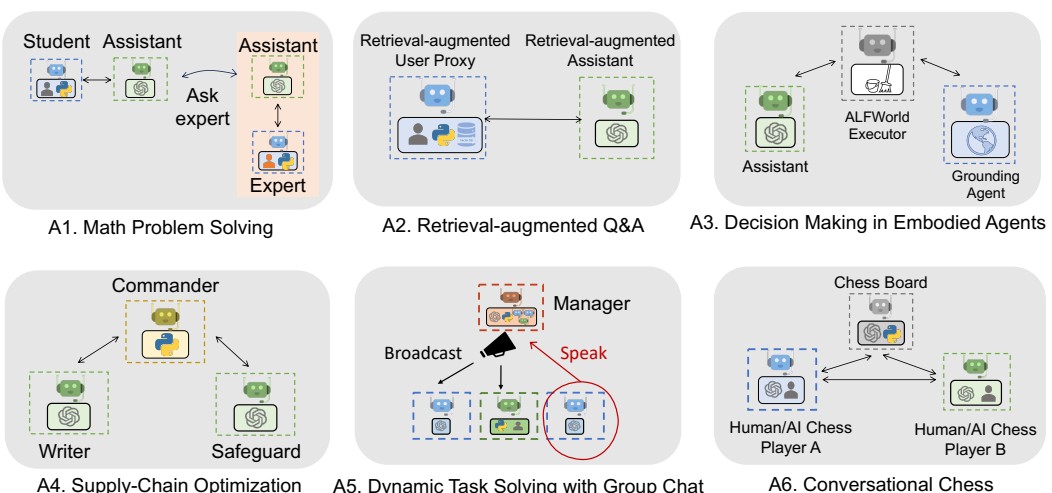

Figure 3: Six examples of applications built using `AutoGen`. These applications demonstrates `AutoGen`'s flexibility in supporting diverse applications with flexible conversation patterns.

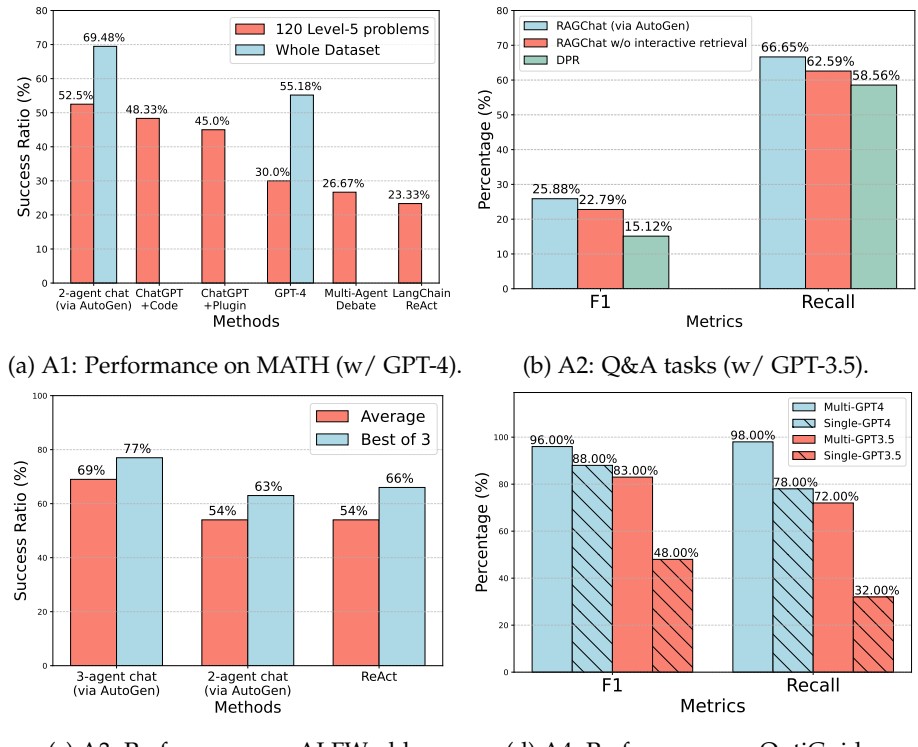

(a) A1: Performance on MATH (w/ GPT-4).

(b) A2: Q&A tasks (w/ GPT-3.5).

(c) A3: Performance on ALFWorld.

(d) A4: Performance on OptiGuide.

Figure 4: Performance on four applications A1-A4. (a) shows that a simple two-agent system with built-in agents from `AutoGen` can be used out of the box to achieve the most competitive performance on math problem solving tasks; (b) shows that `AutoGen` can be used to realize effective retrieval augmentation and realize a novel interactive retrieval feature to boost performance on Q&A tasks; (c) shows that `AutoGen` can be used to introduce a three-agent system with a grounding agent to improve performance on ALFWorld; (d) shows that a multi-agent design is helpful in boosting performance in coding tasks that need safeguards.

**A1: Math Problem Solving**

Mathematics is a foundational discipline and the promise of leveraging LLMs to assist with math problem solving opens up a new plethora of applications and avenues for exploration, including personalized AI tutoring, AI research assistance, etc. This section demonstrates how `AutoGen` can help develop LLM applications for math problem solving, showcasing strong performance and flexibility in supporting various problem-solving paradigms.

(**Scenario 1**) We are able to build a system for autonomous math problem solving by directly reusing two built-in agents from `AutoGen`. We evaluate our system and several alternative approaches, including open-source methods such as Multi-Agent Debate (Liang et al., 2023), LangChain ReAct (LangChain, 2023), vanilla GPT-4, and commercial products ChatGPT + Code Interpreter, and ChatGPT + Plugin (Wolfram Alpha), on the MATH (Hendrycks et al., 2021) dataset and summarize the results in Figure 4a. We perform evaluations over 120 randomly selected level-5 problems and on the entire[3] test dataset from MATH. The results show that the built-in agents from `AutoGen` already yield better performance out of the box compared to the alternative approaches, even including the commercial ones. (**Scenario 2**) We also showcase a human-in-the-loop problem-solving process with the help of `AutoGen`. To incorporate human feedback with `AutoGen`, one only needs to set `human_input_mode=`'ALWAYS' in the `UserProxyAgent` of the system in scenario 1. We demonstrate that this system can

---

[3]We did not evaluate ChatGPT on the whole dataset since it requires substantial manual effort and is restricted by its hourly message-number limitation. Multi-agent debate and LangChain ReAct were also not evaluated since they underperformed vanilla GPT-4 on the smaller test set.

effectively incorporate human inputs to solve challenging problems that cannot be solved without humans. (**Scenario 3**) We further demonstrate a novel scenario where *multiple* human users can participate in the conversations during the problem-solving process. Our experiments and case studies for these scenarios show that AutoGen enables better performance or new experience compared to other solutions we experimented with. Details of the evaluation, including case studies in three scenarios are in Appendix E.

## A2: Retrieval-Augmented Q&A

Retrieval augmentation generation (RAG) has emerged as a practical and effective approach for mitigating the intrinsic limitations of LLMs by incorporating external documents. We employ AutoGen to build a RAG system (Lewis et al., 2020; Parvez et al., 2021) named Retrieval-augmented Chat. The system consists of two agents: a Retrieval-augmented User Proxy agent and a Retrieval-augmented Assistant agent, both of which are extended from built-in agents from AutoGen. The Retrieval-augmented User Proxy includes a vector database (Chroma, 2023) as the context retriever. A detailed workflow description of the Retrieval-augmented Chat (RAGChat) is provided in Appendix E. We evaluate Retrieval-augmented Chat in both question-answering and code-generation scenarios. (**Scenario 1**) We first perform an evaluation regarding natural question answering on the Natural Questions dataset (Kwiatkowski et al., 2019) and report results in Figure 4b. In this evaluation, we compare our system with DPR (Dense Passage Retrieval) following an existing evaluation practice (Adlakha et al., 2023). Leveraging the conversational design and natural-language control, AutoGen introduces a novel *interactive retrieval* feature in this application: whenever the retrieved context does not contain the information, instead of terminating, the LLM-based assistant would reply "*Sorry, I cannot find any information about... UPDATE CONTEXT.*" which will invoke more retrieval attempts. We conduct an ablation study in which we prompt the assistant agent to say "*I don't know*" instead of "*UPDATE CONTEXT.*" in cases where relevant information is not found, and report results in Figure 4b. The results show that the interactive retrieval mechanism indeed plays a non-trivial role in the process. We give a concrete example and results using this appealing feature in Appendix E. (**Scenario 2**) We further demonstrate how Retrieval-augmented Chat aids in generating code based on a given codebase that contains code not included in GPT-4's training data. Evaluation and demonstration details for both scenarios are included in Appendix E.

## A3: Decision Making in Embodied Agents

In this subsection, we demonstrate how AutoGen can be used to develop effective applications that involve interactive or online decision-making. We perform the study using the ALFWorld (Shridhar et al., 2021), a diverse collection of synthetic language-based interactive decision-making tasks in household environments. With AutoGen, we implemented a two-agent system for ALFWorld. It consists of an LLM-backed assistant agent for suggesting plans to conduct a task and an executor agent for executing actions. This system integrates the ReAct prompting (Yao et al., 2022) and could achieve similar performance. A common challenge encountered in both ReAct and the AutoGen-based two-agent system is their occasional inability to follow basic commonsense knowledge about the physical world, therefore getting stuck with repetitive errors. Fortunately, the modular design of AutoGen allows us to address this issue effectively: With AutoGen, we introduce a grounding agent, which supplies crucial commonsense knowledge–such as "*You must find and take the object before you can examine it. You must go to where the target object is before you can use it.*"–whenever the system exhibits early signs of recurring errors. It significantly enhances the system's ability to avoid getting entangled in error loops. We compare the task-solving performance of the two variants of our system with GPT-3.5-turbo and ReAct on the 134 unseen tasks from ALFWorld and report results in Figure 4c. The results show that introducing a grounding agent leads to a 15% performance gain on average. Upon examining the systems' outputs, we observe that the grounding agent, by delivering commonsense knowledge at the right junctures, significantly mitigated the tendency of the system to persist with a flawed plan, thereby avoiding the creation of error loops. For an example trajectory comparing the systems see Appendix E, Figure 10.

**A4: Supply-Chain Optimization**

In this subsection, we use `AutoGen` to build a multi-agent coding system based on OptiGuide (Li et al., 2023a), a system that excels at writing code to interpret optimization solutions and answer user questions, such as exploring the implications of changing a supply-chain decision or understanding why the optimizer made a particular choice. The second sub-figure of Figure 3 shows the `AutoGen`-based implementation. The workflow is as follows: the end user sends questions, such as "*What if we prohibit shipping from supplier 1 to roastery 2?*" to the Commander agent. The Commander coordinates with two assistant agents, including the Writer and the Safeguard, to answer the question. The Writer is responsible for crafting code and the Safeguard is responsible for checking the code safety. With `AutoGen` the core workflow code for OptiGuide was reduced from over 430 lines to 100 lines, leading to significant productivity improvement. We provide a detailed comparison of user experience with ChatGPT+Code Interpreter and `AutoGen`-based OptiGuide in Appendix E, where we show that `AutoGen`-based OptiGuide could save around 3x of user's time and reduce user interactions by 3 - 5 times on average. We also conduct an ablation showing that multi-agent abstraction is necessary. Specifically, we construct a single-agent approach where a single agent conducts both the code-writing and safeguard processes. We tested the single- and multi-agent approaches on a dataset of 100 coding tasks, which is crafted to include equal numbers of safe and unsafe tasks. Evaluation results as reported in Figure 4d show that the multi-agent design boosts the F-1 score in identifying unsafe code by 8% (with GPT-4) and 35% (with GPT-3.5-turbo), and the recall by 20% (with GPT-4) and 40% (with GPT-3.5-turbo).

**A5: Dynamic Task Solving with Group Chat**

There are many real-world scenarios that require dynamic task solving, meaning that the exact workflow cannot be predetermined due to the complex nature of the tasks. For example, tasks involving coding and web scraping. The *group chat* conversation pattern in `AutoGen` is a desirable pattern in such scenarios. In a group chat, the participating agents share the same context and converse with the others in a dynamic manner instead of following a pre-defined order. Dynamic group chat relies on ongoing conversations to guide the flow of interaction among agents. These make dynamic group chat ideal for situations where collaboration without strict communication order is beneficial. In `AutoGen`, the `GroupChatManager` class serves as the conductor of conversation among agents and repeats the following three steps: dynamically selecting a speaker, collecting responses from the selected speaker, and broadcasting the message (Figure 3-A5). For the dynamic speaker-selection component, we use a role-play style prompt. Through a pilot study on 12 manually crafted complex tasks, we observed that compared to a prompt that is purely based on the task, utilizing a role-play prompt often leads to more effective consideration of both conversation context and role alignment during the problem-solving and speaker-selection process. Consequently, this leads to a higher success rate and fewer LLM calls. We include detailed results in Appendix E.

**A6: Conversational Chess**

Using `AutoGen`, we developed Conversational Chess, a natural language interface game shown in the last sub-figure of Figure 3. It features built-in agents for players, which can be human or LLM, and a third-party board agent to provide information and validate moves based on standard rules. With `AutoGen`, we enabled two essential features: (1) Natural, flexible, and engaging game dynamics, enabled by the customizable agent design in `AutoGen`. Conversational Chess supports a range of game-play patterns, including AI-AI, AI-human, and human-human, with seamless switching between these modes during a single game. An illustrative example of these entertaining game dynamics can be found in Figure 15, Appendix E. (2) Grounding, which is a crucial aspect to maintain game integrity. During gameplay, the board agent checks each proposed move for legality; if a move is invalid, the agent responds with an error, prompting the player agent to re-propose a legal move before continuing. This process ensures that only valid moves are played and helps maintain a consistent gaming experience. As an ablation study, we removed the board agent and instead only relied on a relevant prompt "*you should make sure both you and the opponent*

*are making legal moves"* to ground their move. The results highlighted that without the board agent, illegitimate moves caused game disruptions. The modular design offered flexibility, allowing swift adjustments to the board agent in response to evolving game rules or varying chess rule variants. A comprehensive demonstration of this ablation study is in Appendix E.

## 4 Discussion

We introduced an open-source library, `AutoGen`, that incorporates paradigms of conversable agents and conversation programming. `AutoGen` also provides various additional supports, including multimodality, asynchronous operations, and enhanced LLM inference. Furthermore, `AutoGen` seamlessly interoperates with numerous single-agent systems, LLM tools, and libraries, such as OpenAI Assistant and MemGPT (Packer et al., 2023). Although still in an early stage, `AutoGen` is already benefiting a wide range of vertical industries and empowering researchers to build multi-agent AI systems for various scientific studies. For example, `AutoGen` has been used to realize a multi-agent system for accessing task utility in LLM-powered applications (Arabzadeh et al., 2024). AutoGen has been used in studying behaviors of embodied agents in organized teams (Guo et al., 2024). AutoGen is used for producing synthetic dataset for language model fine tuning (Mitra et al., 2024), or in RL environments to train LLMs for agents (Zhou et al., 2024). AutoGen is also used in diverse science and engineering domains such as mechanics (Ni & Buehler, 2023), protein discovery (Ghafarollahi & Buehler, 2024a), and material design (Ghafarollahi & Buehler, 2024b).

`AutoGen` also paves the way for numerous future directions and research opportunities. For instance, it is worth investigating which strategies, such as agent topology and conversation patterns, lead to the most effective multi-agent conversations while optimizing the overall efficiency, among other factors. While increasing the number of agents and other degrees of freedom presents opportunities for tackling more complex problems, it may also introduce new safety challenges that require additional studies and careful consideration. We consider it important future work to explore those safety implications. We provide an expanded discussion in Appendix C, including guidelines for using `AutoGen` and future work. We welcome contributions from the broader community.

## Acknowledgments

The work presented was made possible through discussions with and feedback from a large number of colleagues, including Peter Lee, Johannes Gehrke, Eric Horvitz, Steven Lucco, Umesh Madan, Robin Moeur, Piali Choudhury, Saleema Amershi, Adam Fourney, Victor Dibia, Guoqing Zheng, Corby Rosset, Ricky Loynd, Ece Kamar, Rafah Hosn, John Langford, Ida Momennejad, Brian Krabach, Taylor Webb, Shanka Subhra Mondal, Wei-ge Chen, Robert Gruen, Yinan Li, Yue Wang, Suman Nath, Tanakorn Leesatapornwongsa, Xin Wang, Shishir Patil, Tianjun Zhang, Saehan Jo, Ishai Menache, Kontantina Mellou, Runlong Zhou, Feiran Jia, Hamed Khanpour, Hamid Palangi, Srinagesh Sharma, Julio Albinati Cortez, Amin Saied, Yuzhe Ma, Dujian Ding, Linyong Nan, Prateek Yadav, Shannon Shen, Ankur Mallick, Mark Encarnación, Lars Liden, Tianwei Yue, Julia Kiseleva, Anastasia Razdaibiedina, and Luciano Del Corro. Li Jiang would like to thank Jeff Zheng and DJ Lan for their support during this work. Authors would like to especially thank many colleagues from Microsoft Research AI Frontiers who contributed substantially to AutoGen's growth, key new features, and reliability, especially Jack Gerrits, Adam Fourney, Victor Dibia, Olga Vrousgou, Rajan Chari, Cheng Tan, and Ricky Loynd. Last but not least, we would like to thank individuals from the open-source community, including but not limited to Davor Runje, Hrushikesh Dokala, Mark Sze, Wael Karkoub, Yulong Zhai, Joshua Kim, Olaoluwa Ademola Salami, Rajan Chari, Maxim Saplin, Ikko Eltociear Ashimine, Aaron Ward, Linxin Song, David Luong, Zvi Baratz, Yixuan Zhai, among others for contributing to and helping maintain the open-source project.

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

## Ethics statement

There are several potential ethical considerations that could arise from the development and use of the `AutoGen` framework.

- Privacy and Data Protection: The framework allows for human participation in conversations between agents. It is important to ensure that user data and conversations are protected, and that developers use appropriate measures to safeguard privacy.
- Bias and Fairness: LLMs have been shown to exhibit biases present in their training data (Navigli et al., 2023). When using LLMs in the `AutoGen` framework, it is crucial to address and mitigate any biases that may arise in the conversations between agents. Developers should be aware of potential biases and take steps to ensure fairness and inclusivity.
- Accountability and Transparency: As discussed in the future work section, as the framework involves multiple agents conversing and cooperating, it is important to establish clear accountability and transparency mechanisms. Users should be able to understand and trace the decision-making process of the agents involved in order to ensure accountability and address any potential issues or biases.
- Trust and Reliance: `AutoGen` leverages human understanding and intelligence while providing automation through conversations between agents. It is important to consider the impact of this interaction on user experience, trust, and reliance on AI systems. Clear communication and user education about the capabilities and limitations of the system will be essential (Cai et al., 2019).
- Unintended Consequences: As discussed before, the use of multi-agent conversations and automation in complex tasks may have unintended consequences. Especially, allowing LLM agents to make changes in external environments through code execution or function calls, such as install packages, could be risky. Developers should carefully consider the potential risks and ensure that appropriate safeguards are in place to prevent harm or negative outcomes.

## Reproducibility Statement

Following the best practices in the community, we aim to ensure the reproducibility of the results presented in this paper. The appendices and the accompanying repository are designed to provide readers and reviewers with comprehensive resources to understand, replicate, and extend our work.

- Detailed descriptions of each application, including their construction and evaluations, are provided in Appendix E.
- Example outputs from `AutoGen` and alternative approaches across different applications are illustrated in Appendix F.
- A repository containing the source code of `AutoGen` (v0.1.1) and application evaluation code is provided, along with instructions for reproducing the experiments reported in this paper: `https://github.com/qingyun-wu/autogen/tree/2024-03`.

## A   Interfaces and Example Code

**Agent Specification.** `AutoGen` provides a set of built-in and pre-configured conversable agents, which can be directly imported and used. One can further create an agent with customized capabilities and proper configuration when instantiating the agent.

```
from autogen import ConversableAgent, AssistantAgent, UserProxyAgent
user_proxy = UserProxyAgent(name="user proxy")
```

Listing 1: Importing and using pre-configured agent classes from AutoGen

```
my_assistant = ConversableAgent(
    name="my assistant",
    llm_config=my_config,
    code_execution_config=True,
    function_map=my_func,
)
```

Listing 2: Specifying an agent with customized capabilities

**High-level agent interfaces for composing diverse conversation patterns.** `AutoGen` provides several commonly used conversation patterns via the following interfaces:

- `initiate_chat`: two-agent chat, which is a chat involving two agents. An example is shown in Listing 4.

- `initiate_chats`: sequential chat, which involves a sequence of chats between two agents. This approach is particularly beneficial for tasks requiring a sequence of interdependent multi-agent conversations. The utility of this method is exemplified in Listing 5, showcasing the execution of complex tasks through coordinated chat sequences.

- `register_nested_chat`: nested chat, which is a chat created by a receiver agent after receiving a message from a sender agent and finished before the receiver agent replies to this message. Nested chats allow one agent to use other agents as their inner monologue to accomplish tasks. This abstraction is powerful as it allows one to compose agents in rich ways. Listing 3 gives an example in which we use this interface to nest a reflection agent within the user_proxy agent. This could be used to realize the self-reflection idea (Shinn et al., 2023) to improve LLM's reasoning and problem-solving capability.

- `GroupChat`: group chat, which is a pattern in which participating agents have shared context. It is desirable for dynamic task solving, as detailed in A5.

```
user_proxy.register_nested_chat([{"recipient": critique_agent}])
```

Listing 3: Nested chat registration (assuming all the agent involved have been created)

```
user_proxy.initiate_chat(
    my_assistant, message="Plot a chart of META and TESLA stock price
    change YTD."
)
```

Listing 4: Two-agent chat (with a nested chat if the corresponding registration is done)

```
initiate_chats(
    [{
            "sender": user_proxy,
            "recipient": my_assistant_1,
            "message": "Get META and TESLA stock price change YTD.",
    },
    {
            "sender": user_proxy,
            "recipient": my_assistant_2,
            "message": "Plot a chart based on the stock price data.",
    }]
)
```

Listing 5: Sequential chat

# B  Expanded Related Work

Here we examine existing LLM-based agent systems or frameworks that can be used to build LLM applications in detail. We categorize the related work into single-agent and multi-agent systems and specifically provide a summary of differentiators comparing `AutoGen` with existing multi-agent systems below. Note that many of these systems are evolving open-source projects, so the remarks and statements about them may only be accurate as of the time of writing. We refer interested readers to detailed LLM-based agent surveys (Xi et al., 2023; Wang et al., 2023)

**Single-Agent Systems:**

- **AutoGPT**: AutoGPT is an open-source implementation of an AI agent that attempts to autonomously achieve a given goal (AutoGPT, 2023). It follows a single-agent paradigm in which it augments the AI model with many useful tools, and does not support multi-agent collaboration.
- **ChatGPT+ (with code interpreter or plugin)**: ChatGPT, a conversational AI service or agent, can now be used alongside a code interpreter or plugin (currently available only under the premium subscription plan ChatGPT Plus) (OpenAI, 2023). The code interpreter enables ChatGPT to execute code, while the plugin enhances ChatGPT with a wide range of curated tools.
- **LangChain Agents**: LangChain is a general framework for developing LLM-based applications (LangChain, 2023). LangChain Agents is a subpackage for using an LLM to choose a sequence of actions. There are various types of agents in LangChain Agents, with the ReAct agent being a notable example that combines reasoning and acting when using LLMs (mainly designed for LLMs prior to ChatGPT) (Yao et al., 2022). All agents provided in LangChain Agents follow a single-agent paradigm and are not inherently designed for communicative and collaborative modes. A significant summary of its limitations can be found in (Woolf, 2023). Due to these limitations, even the multi-agent systems in LangChain (e.g., re-implementation of CAMEL) are not based on LangChain Agents but are implemented from scratch. Their connection to LangChain lies in the use of basic orchestration modules provided by LangChain, such as AI models wrapped by LangChain and the corresponding interface.
- **Transformers Agent**: Transformers Agent (HuggingFace, 2023) is an experimental natural-language API built on the transformers repository. It includes a set of curated tools and an agent to interpret natural language and use these tools. Similar to AutoGPT, it follows a single-agent paradigm and does not support agent collaboration.

`AutoGen` differs from the single-agent systems by supporting multi-agent conversations.

**Multi-Agent Systems:**

- **Multi-Agent Debate:** Two recent works investigate and show that multi-agent debate is an effective way to encourage divergent thinking in LLMs (Liang et al., 2023) and to improve the factuality and reasoning of LLMs (Du et al., 2023). In both works, multiple LLM inference instances are constructed as multiple agents to solve problems with agent debate. Each agent is simply an LLM inference instance, while no tool or human is involved, and the inter-agent conversation needs to follow a pre-defined order. These works attempt to build LLM applications with multi-agent conversation, while `AutoGen`, designed as a generic infrastructure, can be used to facilitate this development and enable more applications with dynamic conversation patterns.
- **MetaGPT** (Hong et al., 2023) and **ChatDev** (Qian et al., 2023): Both MetaGPT and ChatDev are multi-agent frameworks for software development. Compared to MetaGPT and ChatDev, `AutoGen` supports more flexible and complex workflows in addition to Standardized Operating Procedures (SOP) or chains, e.g., nested chat and group chat, and covers a wider range of usage scenarios beyond software engineering.
- **Simulacra** (Park et al., 2023): This system features twenty-five generative agents in an interactive sandbox environment. It represents one of the initial attempts at multi-agent

interaction, with a special focus on investigating whether generative agents could enable believable simulations of human behavior.

- **BabyAGI**: BabyAGI (BabyAGI, 2023) is an example implementation of an AI-powered task management system in a Python script. In this implemented system, multiple LLM-based agents are used. For example, there is an agent for creating new tasks based on the objective and the result of the previous task, an agent for prioritizing the task list, and an agent for completing tasks/sub-tasks. As a multi-agent system, BabyAGI adopts a static agent conversation pattern, i.e., a predefined order of agent communication, while `AutoGen` supports both static and dynamic conversation patterns and additionally supports tool usage and human involvement.

- **CAMEL**: CAMEL (Li et al., 2023b) is a role-playing-based agent framework supporting two or three agents (depending on if a critic agent is included). It demonstrates how role-playing can be used to let chat agents communicate with each other for task completion. It also records agent conversations for behavior analysis and capability understanding. An Inception-prompting technique is used to achieve autonomous cooperation between agents.

Several other LLM-based multi-agent systems/frameworks have emerged following the initial release of our work, including AgentVerse (Chen et al., 2023b), AutoAgents (Chen et al., 2023a), OpenAgents (Xie et al., 2023) and IoA (Chen et al., 2024). We omit a detailed discussion here due to the contemporaneous nature of these developments.

## C Expanded Discussion

The applications in Section 3 show how `AutoGen` not only enables new applications but also helps renovate existing ones. For example, in A1 (scenario 3), A5, and A6, `AutoGen` enabled the creation of multi-agent conversations that follow a dynamic pattern instead of a fixed back-and-forth. And in both A5 and A6, humans can participate in the activities together with multiple other AI agents in a conversational manner. Similarly, A1-A4 show how popular applications can be renovated quickly with `AutoGen`. Despite the complexity of these applications (most of them involve more than two agents or dynamic multi-turn agent cooperation), our `AutoGen`-based implementation remains simple, demonstrating promising opportunities to build creative applications and a large space for innovation. In reflecting on *why* these benefits can be achieved in these applications with `AutoGen`, we believe there are a few reasons:

- **Ease of use**: The built-in agents can be used out-of-the-box, delivering strong performance even without any customization. (A1, A3)

- **Modularity**: The division of tasks into separate agents promotes modularity in the system. Each agent can be developed, tested, and maintained independently, simplifying the overall development process and facilitating code management. (A3, A4, A5, and A6)

- **Programmability:** `AutoGen` allows users to extend/customize existing agents to develop systems satisfying their specific needs with ease. (A1-A6). As demonstrated in A4, the use of `AutoGen` significantly streamlines the core workflow code, reducing it from over 430 lines to just 100 lines, resulting in a fourfold reduction in code size.

- **Allowing human involvement**: `AutoGen` provides a native mechanism to achieve human participation and/or human oversight. With `AutoGen`, humans can seamlessly and optionally cooperate with AIs to solve problems or generally participate in the activity. `AutoGen` also facilitates interactive user instructions to ensure the process stays on the desired path. (A1, A2, A5, and A6)

- **Collaborative/adversarial agent interactions**: Like many collaborative agent systems (Dong et al., 2023), agents in `AutoGen` can share information and knowledge, to complement each other's abilities and collectively arrive at better solutions. (A1, A2, A3, and A4). Analogously, in certain scenarios, some agents are required to work in an adversarial way. Relevant information is shared among different conversations in a controlled manner, preventing distraction or hallucination. (A4, A6). `AutoGen` supports both patterns, enabling effective utilization and augmentation of LLMs.

### C.1 General Guidelines for Using `AutoGen`

Below we give some recommendations for using agents in `AutoGen` to accomplish a task.

1. **Consider using built-in agents first.** For example, `AssistantAgent` is pre-configured with a carefully designed system message for generic problem-solving via code. The `UserProxyAgent` is configured to solicit human inputs and perform tool execution. Many problems can be solved by simply combining these two agents. When customizing agents for an application, consider the following options: (1) human input mode, termination condition, code execution configuration, and LLM configuration can be specified when constructing an agent; (2) `AutoGen` supports adding instructions in an initial user message, which is an effective way to boost performance without needing to modify the system message; (3) `UserProxyAgent` can be extended to handle different execution environments and exceptions, etc.; (4) when system message modification is needed, consider leveraging the LLM's capability to program its conversation flow with natural language.

2. **Start with a simple conversation topology**. Consider using the two-agent chat or the group chat setup first, as they can often be extended with the least code. Note that the two-agent chat can be easily extended to involve more than two agents by using LLM-consumable functions in a dynamic way.

3. Try to **reuse built-in reply methods** based on LLM, tool, or human before implementing a custom reply method because they can often be reused to achieve the goal in a simple way (e.g., the built-in agent `GroupChatManager`'s reply method reuses the built-in LLM-based reply function when selecting the next speaker, ref. A5 in Section 3).

4. When developing a new application with `UserProxyAgent`, **start with humans always in the loop**, i.e., human_input_mode='ALWAYS', even if the target operation mode is more autonomous. This helps evaluate the effectiveness of `AssistantAgent`, tuning the prompt, discovering corner cases, and debugging. Once confident with small-scale success, consider setting human_input_mode = 'NEVER'. This enables LLM as a backend, and one can either use the LLM or manually generate diverse system messages to simulate different use cases.

5. Despite the numerous advantages of `AutoGen` agents, there could be cases/scenarios where **other libraries/packages could help**. For example: (1) For (sub)tasks that do not have requirements for back-and-forth trouble-shooting, multi-agent interaction, etc., a unidirectional (no back-and-forth message exchange) pipeline can also be orchestrated with LangChain (LangChain, 2023), LlamaIndex (Liu, 2022), Guidance (Guidance, 2023), Semantic Kernel (SemanticKernel, 2023), Gorilla (Patil et al., 2023) or low-level inference API ('autogen.oai' provides an enhanced LLM inference layer at this level) (Dibia, 2023). (2) When existing tools from LangChain etc. are helpful, one can use them as tool backends for `AutoGen` agents. For example, one can readily use tools, e.g., Wolfram Alpha, from LangChain in `AutoGen` agent. (3) For specific applications, one may want to leverage agents implemented in other libraries/packages. To achieve this, one could wrap those agents as conversable agents in `AutoGen` and then use them to build LLM applications through multi-agent conversation. (4) It can be hard to find an optimal operating point among many tunable choices, such as the LLM inference configuration. Blackbox optimization packages like 'flaml.tune' (Wang et al., 2021; Zhang et al., 2023) can be used together with `AutoGen` to automate such tuning.

### C.2 Future Work

This work raises many research questions and future directions.

**Designing optimal multi-agent workflows:** Creating a multi-agent workflow for a given task can involve many decisions, e.g., how many agents to include (Qian et al., 2024), how to assign agent roles and agent capabilities (Song et al., 2024), how the agents should interact with each other (Wu et al., 2024), and whether to automate a particular part of the workflow. There may not exist a one-size-fits-all answer, and the best solution might depend on the specific application. This raises important questions: For what types of tasks

and applications are multi-agent workflows most useful? How do multiple agents help in different applications? For a given task, what is the optimal (e.g., cost-effective) multi-agent workflow?

**Creating highly capable agents:** `AutoGen` can enable the development of highly capable agents that leverage the strengths of LLMs, tools, and humans. Creating such agents is crucial to ensuring that a multi-agent workflow can effectively troubleshoot and make progress on a task. We believe that more systematic work will be required to develop guidelines for application-specific agents, to create a large OSS knowledge base of agents, and to create agents that can discover and upgrade their skills (Cai et al., 2023; Zhang et al., 2024; Song et al., 2024).

**Enabling scale, safety, and human agency:** Section 3 shows how complex multi-agent workflows can enable new applications, and future work will be needed to assess whether scaling further can help solve extremely complex tasks. However, as these workflows scale and grow more complex, it may become difficult to log and adjust them. Thus, it will become essential to develop clear mechanisms and tools to track and debug their behavior. Otherwise, these techniques risk resulting in incomprehensible, unintelligible chatter among agents (Lewis et al., 2017).

Our work also shows how complex, fully autonomous workflows with `AutoGen` can be useful, but fully autonomous agent conversations will need to be used with care. While the autonomous mode `AutoGen` supports could be desirable in many scenarios, a high level of autonomy can also pose potential risks, especially in high-risk applications (Amodei et al., 2016; Weld & Etzioni, 1994). As a result, building fail-safes against cascading failures and exploitation, mitigating reward hacking, out of control and undesired behaviors, maintaining effective human oversight of applications built with `AutoGen` agents will become important. While `AutoGen` provides convenient and seamless involvement of humans through a user proxy agent, developers and stakeholders still need to understand and determine the appropriate level and pattern of human involvement to ensure the safe and ethical use of the technology (Horvitz, 1999; Amershi et al., 2019).

# D    Default System Message for Assistant Agent

Figure 5 shows the default system message for the built-in assistant agent in `AutoGen` (v0.1.1), where we introduce several new prompting techniques and highlight them accordingly. When combining these new prompting techniques together, we can program a fairly complex conversation even with the simplest two-agent conversation topology. This approach tries to exploit the capability of LLMs in implicit state inference to a large degree. LLMs do not follow all the instructions perfectly, so the system design requires additional mechanisms to manage exceptions and faults. Certain instructions can contain ambiguities, and the designer should either mitigate these ambiguities to enhance precision or intentionally retain them to allow for flexibility and address the varying situations through other means.

# E    Application Details

**A1: Math Problem Solving**

**Scenario 1: Autonomous Problem Solving.**    We perform both qualitative and quantitative evaluations in this scenario. For all evaluations, we use GPT-4 as the base model, and pre-install the "sympy" package in the execution environment. We compare `AutoGen` with the following LLM-based agent systems:

- AutoGPT: The out-of-box AutoGPT is used. We initialize AutoGPT by setting the purpose to "solve math problems", resulting in a "MathSolverGPT" with auto-generated goals.
- ChatGPT+Plugin: We enable the Wolfram Alpha plugin (a math computation engine) in the OpenAI web client.

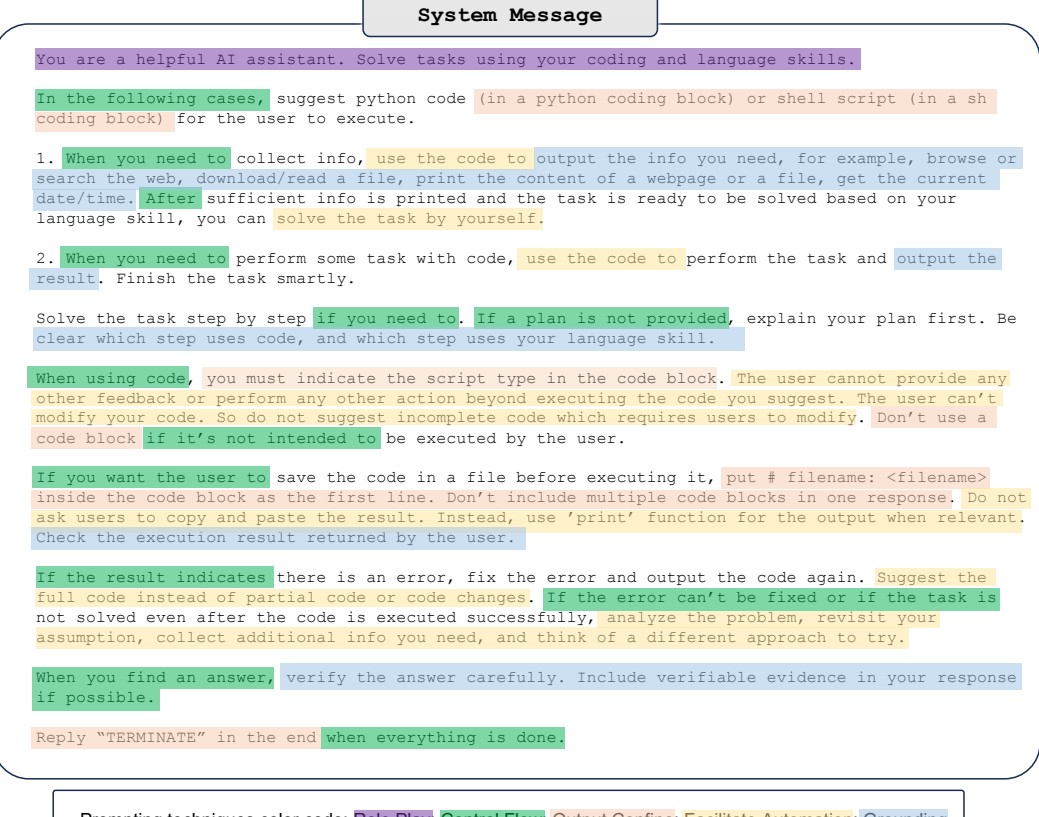

Figure 5: Default system message for the built-in assistant agent in AutoGen (v0.1.1). This is an example of conversation programming via natural language. It contains instructions of different types, including role play, control flow, output confine, facilitate automation, and grounding.

- ChatGPT+Code Interpreter: This is a recent feature in OpenAI web client. Note that the above two premium features from ChatGPT require a paid subscription to be accessed and are the most competitive commercial systems.
- LangChain ReAct+Python: We use Python agent from LangChain. To handle parsing errors, we set "handle_parsing_errors=True", and use the default zero-shot ReAct prompt.
- Multi-Agent Debate (Liang et al., 2023): We modified the code of the multi-agent debate to perform evaluation. By default, there are three agents: an affirmative agent, a negative agent, and a moderator.

We also conducted preliminary evaluations on several other multi-agent systems, including BabyAGI, CAMEL, and MetaGPT. The results indicate that they are not suitable choices for solving math problems out of the box. For instance, when MetaGPT is tasked with solving a math problem, it begins developing software to address the problem, but most of the time, it does not actually solve the problem. We have included the test examples in Appendix F.

For the qualitative evaluation, we utilize two level-5 problems from the MATH dataset, testing each problem three times. The first problem involves simplifying a square root fraction, and the second problem involves solving a number theory issue. The correctness counts and reasons for failure are detailed in Table 1. For the quantitative evaluation, we conduct two sets of experiments on the MATH dataset to assess the correctness of these systems: (1) an experiment involving 120 level-5 (the most challenging level) problems, including 20 problems from six categories, excluding geometry, and (2) an experiment on the entire test set, which includes 5000 problems. We exclude AutoGPT from this evaluation

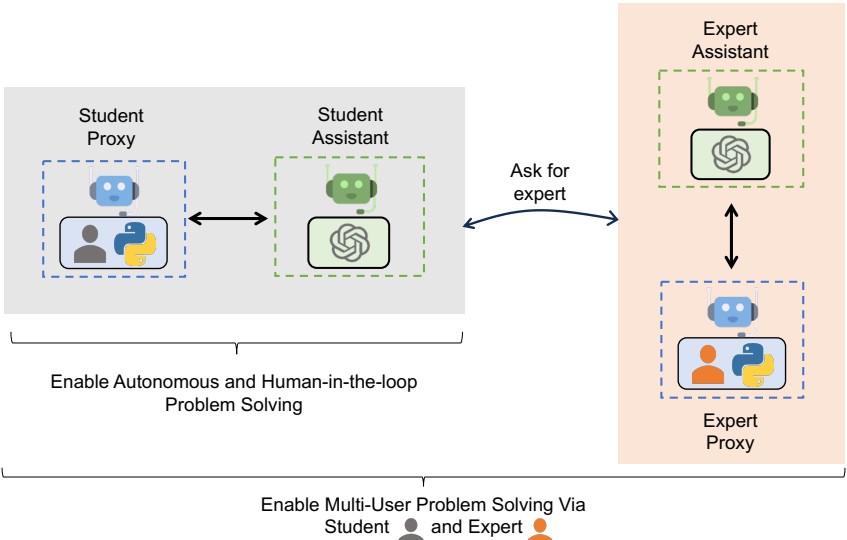

Figure 6: Illustration of three problem-solving scenarios with AutoGen: (Gray) Enables a workflow where a student collaborates with an assistant agent to solve problems, either autonomously or in a human-in-the-loop mode. (Gray + Orange) Facilitates a more sophisticated workflow wherein the assistant, on the fly, can engage another user termed "expert", who is in the loop with their own assistant agent, to aid in problem-solving if its own solutions are not satisfactory.

as it cannot access results from code executions and does not solve any problems in the qualitative evaluation. Our analysis of the entire dataset reveals that AutoGen achieves an overall accuracy of 69.48%, while GPT-4's accuracy stands at 55.18%. From these evaluations, we have the following observations regarding the problem-solving success rate and user experience of these systems:

- Problem-solving success rate: Results from the quantitative evaluations show that AutoGen can help achieve the highest problem-solving success rate among all the compared methods. The qualitative evaluations elucidate common failure reasons across several alternative approaches. ChatGPT+Code Interpreter fails to solve the second problem, and ChatGPT+Plugin struggles to solve both problems. AutoGPT fails on both problems due to code execution issues. The LangChain agent also fails on both problems, producing code that results in incorrect answers in all trials.

- Based on the qualitative evaluation, we analyze the user experience concerning the verbosity of the response and the ability of the LLM-based system to run without unexpected behaviors. ChatGPT+Plugin is the least verbose, mainly because Wolfram queries are much shorter than Python code. AutoGen, ChatGPT+Code Interpreter, and LangChain exhibit similar verbosity, although LangChain is slightly more verbose due to more code execution errors. AutoGPT is the most verbose system owing to predefined steps like THOUGHTS, REASONING, and PLAN, which it includes in replies every time. Overall, AutoGen and ChatGPT+Code Interpreter operate smoothly without exceptions. We note the occurrences of undesired behaviors from other LLM-based systems that could affect user experience: AutoGPT consistently outputs code without the print' statement and cannot correct this, requiring the user to run them manually; ChatGPT with Wolfram Alpha plugin has the potential to become stuck in a loop that must be manually stopped; and Langchain ReAct could exit with a parse error, necessitating the passing of a 'handle_parse_error' parameter.

**Scenario 2: Human-in-the-loop Problem Solving.** For challenging problems that these LLM systems cannot solve autonomously, human feedback during the problem-solving process can be helpful. To incorporate human feedback with AutoGen, one can set

Table 1: Qualitative evaluation of two math problems from the MATH dataset within the autonomous problem-solving scenario. Each LLM-based system is tested three times on each of the problems. This table reports the problem-solving correctness and summarizes the reasons for failure.

| | Correctness | Failure Reason |
|---|---|---|
| AutoGen | 3/3 | N/A. |
| AutoGPT | 0/3 | The LLM gives code without the print function so the result is not printed. |
| ChatGPT+Plugin | 1/3 | The return from Wolfram Alpha contains 2 simplified results, including the correct answer, but GPT-4 always chooses the wrong answer. |
| ChatGPT+Code Interpreter | 2/3 | Returns a wrong decimal result. |
| LangChain ReAct | 0/3 | LangChain gives 3 different wrong answers. |
| Multi-Agent Debate | 0/3 | It gives 3 different wrong answers due to calculation errors. |

(a) Evaluation on the first problem that asks to simplify a square root fraction.

| | Correctness | Failure Reason |
|---|---|---|
| AutoGen | 2/3 | The final answer from code execution is wrong. |
| AutoGPT | 0/3 | The LLM gives code without the print function so the result is not printed. |
| ChatGPT+Plugin | 1/3 | For one trial, GPT-4 got stuck because it keeps giving wrong queries and has to be stopped. Another trial simply gives a wrong answer. |
| ChatGPT+Code Interpreter | 0/3 | It gives 3 different wrong answers. |
| LangChain ReAct | 0/3 | LangChain gives 3 different wrong answers. |
| Multi-Agent Debate | 0/3 | It gives 3 different wrong answers. |

(b) Evaluation on the second number theory problem.

human_input_mode=‘ALWAYS’ in the user proxy agent. We select one challenging problem that none of these systems can solve autonomously across three trials. We adhere to the process outlined below to provide human inputs for all the compared methods:

1. Input the problem: Find the equation of the plane which bisects the angle between the planes $3x - 6y + 2z + 5 = 0$ and $4x - 12y + 3z - 3 = 0$, and which contains the point $(-5, -1, -5)$. Enter your answer in the form

$$Ax + By + Cz + D = 0,$$

where $A$, $B$, $C$, $D$ are integers such that $A > 0$ and $\gcd(|A|, |B|, |C|, |D|) = 1$.

2. The response from the system does not solve the problem correctly. We then give a hint to the model: Your idea is not correct. Let's solve this together. Suppose $P = (x, y, z)$ is a point that lies on a plane that bisects the angle, the distance from P to the two planes is the same. Please set up this equation first.

3. We expect the system to give the correct distance equation. Since the equation involves an absolute sign that is hard to solve, we would give the next hint: Consider the two cases to remove the abs sign and get two possible solutions.

4. If the system returns the two possible solutions and doesn't continue to the next step, we give the last hint: Use point (-5,-1,-5) to determine which is correct and give the final answer.

5. Final answer is $\boxed{11x+6y+5z+86=0}$.

We observed that AutoGen consistently solved the problem across all three trials. Chat-GPT+Code Interpreter and ChatGPT+Plugin managed to solve the problem in two out

of three trials, while AutoGPT failed to solve it in all three attempts. In its unsuccessful attempt, ChatGPT+Code Interpreter failed to adhere to human hints. In its failed trial, ChatGPT+Plugin produced an almost correct solution but had a sign discrepancy in the final answer. AutoGPT was unable to yield a correct solution in any of the trials. In one trial, it derived an incorrect distance equation. In the other two trials, the final answer was incorrect due to code execution errors.

**Scenario 3: Multi-User Problem Solving.** Next-generation LLM applications may necessitate the involvement of multiple real users for collectively solving a problem with the assistance of LLMs. We showcase how AutoGen can be leveraged to effortlessly construct such a system. Specifically, building upon scenario 2 mentioned above, we aim to devise a simple system involving two human users: a student and an expert. In this setup, the student interacts with an LLM assistant to address some problems, and the LLM automatically resorts to the expert when necessary.

The overall workflow is as follows: The student chats with the LLM-based assistant agent through a student proxy agent to solve problems. When the assistant cannot solve the problem satisfactorily, or the solution does not match the expectation of the student, it would automatically hold the conversation and call the pre-defined ask_for_expert function via the *function_call* feature of LLM in order to resort to the expert. Specifically, it would automatically produce the initial message for the ask_for_expert function, which could be the statement of the problem or the request to verify the solution to a problem, and the expert is supposed to respond to this message with the help of the expert assistant. After the conversation between the expert and the expert's assistant, the final message would be sent back to the student assistant as the response to the initial message. Then, the student assistant would resume the conversation with the student using the response from the expert for a better solution. A detailed visualization is shown in Figure 6.

With AutoGen, constructing the student/expert proxy agent and the assistant agents is straightforward by reusing the built-in UserProxyAgent and AssistantAgent through appropriate configurations. The only development required involves writing several lines of code for the ask_for_expert function, which then becomes part of the configuration for the assistant. Additionally, it's easy to extend such a system to include more than one expert, with a specific ask_for_expert function for each, or to include multiple student users with a shared expert for consultation.

**A2: Retrieval-Augmented Q&A**

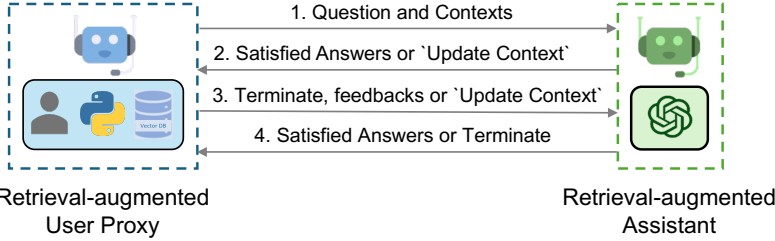

Figure 7: Overview of RAG Chat which involves two agents, including a RAG User Proxy and a Retrieval-augmented Assistant. Given a set of documents, the Retrieval-augmented User Proxy first automatically processes documents—splits, chunks, and stores them in a vector database. Then for a given user input, it retrieves relevant chunks as context and sends it to the Retrieval-augmented Assistant, which uses LLM to generate code or text to answer questions. Agents converse until they find a satisfactory answer.

**Detailed Workflow.** The workflow of RAG Chat is illustrated in Figure 7. To use Retrieval-augmented Chat, one needs to initialize two agents including Retrieval-augmented User Proxy and Retrieval-augmented Assistant. Initializing the Retrieval-Augmented User Proxy necessitates specifying a path to the document collection. Subsequently, the Retrieval-Augmented User Proxy can download the documents, segment them into chunks of a specific size, compute embeddings, and store them in a vector database. Once a chat is initiated, the agents collaboratively engage in code generation or question-answering adhering to the procedures outlined below:

1. The Retrieval-Augmented User Proxy retrieves document chunks based on the embedding similarity, and sends them along with the question to the Retrieval-Augmented Assistant.
2. The Retrieval-Augmented Assistant employs an LLM to generate code or text as answers based on the question and context provided. If the LLM is unable to produce a satisfactory response, it is instructed to reply with "Update Context" to the Retrieval-Augmented User Proxy.
3. If a response includes code blocks, the Retrieval-Augmented User Proxy executes the code and sends the output as feedback. If there are no code blocks or instructions to update the context, it terminates the conversation. Otherwise, it updates the context and forwards the question along with the new context to the Retrieval-Augmented Assistant. Note that if human input solicitation is enabled, individuals can proactively send any feedback, including "Update Context", to the Retrieval-Augmented Assistant.
4. If the Retrieval-Augmented Assistant receives "Update Context", it requests the next most similar chunks of documents as new context from the Retrieval-Augmented User Proxy. Otherwise, it generates new code or text based on the feedback and chat history. If the LLM fails to generate an answer, it replies with "Update Context" again. This process can be repeated several times. The conversation terminates if no more documents are available for the context.

We utilize Retrieval-Augmented Chat in two scenarios. The first scenario aids in generating code based on a given codebase. While LLMs possess strong coding abilities, they are unable to utilize packages or APIs that are not included in their training data, e.g., private codebases, or have trouble using trained ones that are frequently updated post-training. Hence, Retrieval-Augmented Code Generation is considered to be highly valuable. The second scenario involves question-answering on the Natural Questions dataset (Kwiatkowski et al., 2019), enabling us to obtain comparative evaluation metrics for the performance of our system.

**Scenario 1: Evaluation on Natural Questions QA dataset.** In this case, we evaluate the Retrieval-Augmented Chat's end-to-end question-answering performance using the Natural

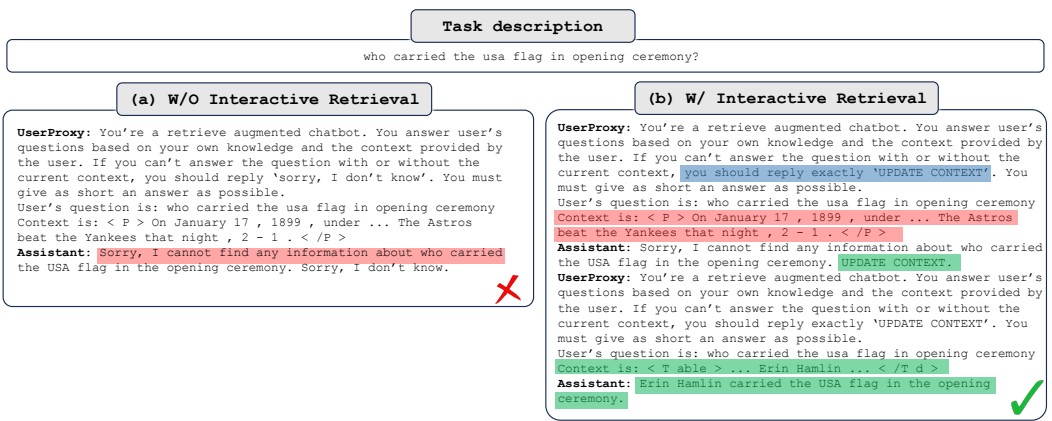

Figure 8: Retrieval-augmented Chat without (W/O) and with (W/) *interactive retrieval*.

**Questions dataset (Kwiatkowski et al., 2019).** We collected 5,332 non-redundant context documents and 6,775 queries from HuggingFace. First, we create a document collection based on the entire context corpus and store it in the vector database. Then, we utilize Retrieval-Augmented Chat to answer the questions. An example (Figure 8) from the NQ dataset showcases the advantages of the *interactive retrieval* feature: *"who carried the usa flag in opening ceremony"*. When attempting to answer this question, the context with the highest similarity to the question embedding does not contain the required information for a response. As a result, the LLM assistant (GPT-3.5-turbo) replies *"Sorry, I cannot find any information about who carried the USA flag in the opening ceremony. UPDATE CONTEXT."* With the unique and innovative ability to update context in Retrieval-Augmented Chat, the user proxy agent automatically updates the context and forwards it to the assistant agent again. Following this process, the agent is able to generate the correct answer to the question.

In addition, we conduct an experiment using the same prompt as illustrated in (Adlakha et al., 2023) to investigate the advantages of AutoGen *W/O interactive retrieval*. The F1 score and Recall for the first 500 questions are 23.40% and 62.60%, respectively, aligning closely with the results reported in Figure 4b. Consequently, we assert that AutoGen *W/O interactive retrieval* outperforms *DPR* due to differences in the retrievers employed. Specifically, we utilize a straightforward vector search retriever with the *all-MiniLM-L6-v2* model for embeddings.

Furthermore, we analyze the number of LLM calls in experiments involving both AutoGen and AutoGen *W/O interactive retrieval*, revealing that approximately 19.4% of questions in the Natural Questions dataset trigger an "Update Context" operation, resulting in additional LLM calls.

**Scenario 2: Code Generation Leveraging Latest APIs from the Codebase.** In this case, the question is *"How can I use FLAML to perform a classification task and use Spark for parallel training? Train for 30 seconds and force cancel jobs if the time limit is reached."*. FLAML (v1) (Wang et al., 2021) is an open-source Python library designed for efficient AutoML and tuning. It was open-sourced in December 2020, and is included in the training data of GPT-4. However, the question necessitates the use of Spark-related APIs, which were added in December 2022 and are not encompassed in the GPT-4 training data[1]. Consequently, the original GPT-4 model is unable to generate the correct code, due to its lack of knowledge regarding Spark-related APIs. Instead, it erroneously creates a non-existent parameter, *spark*, and sets it to True'. Nevertheless, with Retrieval-Augmented Chat, we provide the latest reference documents as context. Then, GPT-4 generates the correct code blocks by setting *use_spark* and *force_cancel* to True'.

---

[1]*gpt-4-0613* is used in this experiment. Its cutoff date is September 2021.

## A3: Decision Making in Embodied Agents

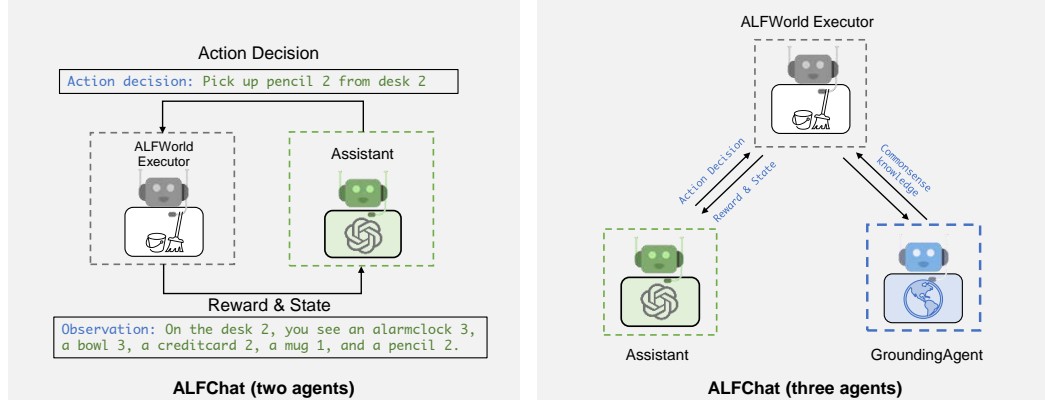

Figure 9: Overview of two versions of ALFChat: a two-agent approach where an assistant agent proposes the subsequent step and an executor carries out actions while providing feedback, and a three-agent design that incorporates a grounding agent to furnish the executor with commonsense facts when required.

ALFWorld (Shridhar et al., 2021) is a synthetic language-based interactive decision-making task. It comprises textual environments that aim to simulate real-world household scenes. Given a high-level goal (e.g., putting a hot apple in the fridge) and the description of the household environment, the agent needs to explore and interact with the simulated household environment through a textual interface. A typical task environment contains various types of locations and could require more than 40 steps to finish, which requires agents to decompose the goal into subtasks and tackle them one by one, while effectively exploring the environments.

**Detailed Workflow.** We first propose a straightforward two-agent system with `AutoGen`, illustrated on the left-hand side of Figure 9, to tackle tasks from this benchmark. The system consists of an assistant agent and an executor agent. The assistant agent generates plans and makes action decisions to solve the tasks. The executor agent is tailored specifically for ALFWorld. It performs actions proposed by the assistant and reports action execution results in the household environment as feedback to the assistant. Due to the strict format requirements for the output format, we use the BLEU metric to evaluate the similarity of the output to all valid action options. The option with the highest similarity will be chosen as the action for this round.

A significant challenge encompassed in ALFWorld involves commonsense reasoning. The agent needs to extract patterns from the few-shot examples provided and combine them with the agent's general knowledge of household environments to fully understand task rules. More often than not, the assistant tends to neglect some basic knowledge of the household environment. Thanks to the easy-to-implement multi-agent conversational feature of `AutoGen`, enhancing the assistant agent's reasoning ability by adding a new grounding agent to provide commonsense facts for the decision-making agent's reference becomes straightforward. By scrutinizing the failed attempts and summarizing the reasons for failure, we obtained a holistic understanding of the commonsense knowledge that the assistant agent lacks. Then, we set a grounding agent to provide this general knowledge when the task begins and whenever the assistant outputs the same action three times in a row. This ensures the assistant takes this commonsense knowledge into consideration.

We compare our system's performance with ReAct. ReAct (Yao et al., 2022) is a few-shot prompting technique that interleaves reasoning and acting, allowing for greater synergy between the two and significantly improving performance on both language and decision-making tasks. We integrate ReAct into `AutoGen` by modifying the prompts into a conver-

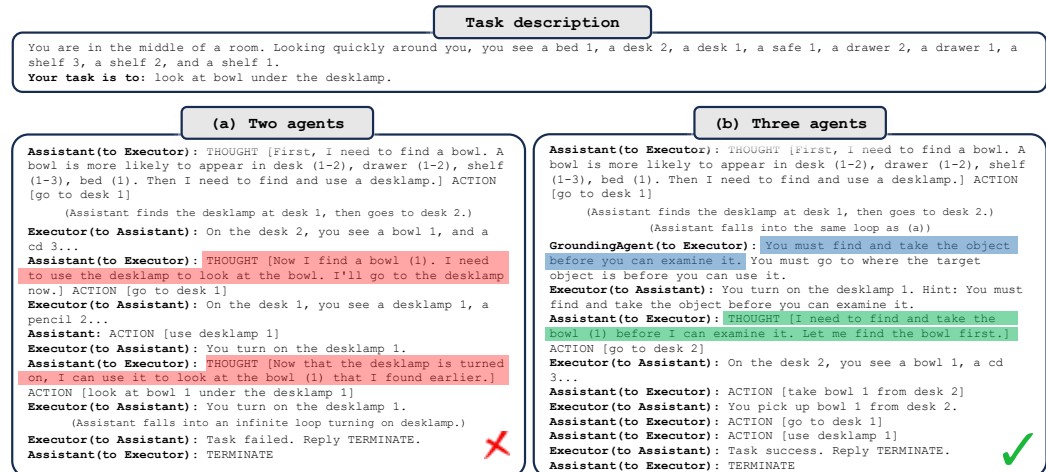

Figure 10: Comparison of results from two designs: (a) two-agent design, which consists of an assistant and an executor, and (b) three-agent design, which adds a grounding agent that serves as a knowledge source. For simplicity, we omit the in-context examples and part of the exploration trajectory, and only show parts contributing to the failure/success of the attempt.

sational manner. Following ReAct, we employ a two-shot setting. The few-shot prompts are obtained from the corresponding repository. As shown in Table 2, the two-agent design matches the performance of ReAct, while the three-agent design significantly outperforms ReAct. On the other hand, introducing a grounding agent as a knowledge source remarkably advances performance on all types of tasks.

**Case study**. Figure 10 exemplifies how a three-agent design eliminates one root cause for failure cases. Most of the tasks involve taking an object and then performing a specific action with it (e.g., finding a vase and placing it on a cupboard). Without a grounding agent, the assistant frequently conflates finding an object with taking it, as illustrated in Figure 10(a). This leads to most of the failure cases in 'pick' and 'look' type tasks. With the introduction of a grounding agent, the assistant can break this loop and successfully complete the task.

**Takeaways.** We introduced a grounding agent to serve as an external commonsense knowledge source, which significantly enhanced the assistant's ability to make informed decisions. This proves that providing necessary commonsense facts to the decision-making agent can assist it in making more informed decisions, thus effectively boosting the task success rate. `AutoGen` brings both simplicity and modularity when adding the grounding agent.

| Method | Pick | Clean | Heat | Cool | Look | Pick 2 | All |
|---|---|---|---|---|---|---|---|
| ReAct (avg) | 63 | 52 | 48 | 71 | 61 | 24 | 54 |
| ALFChat (2 agents)(avg) | 61 | 58 | 57 | 67 | 50 | 19 | 54 |
| ALFChat (3 agents)(avg) | 79 | 64 | 70 | 76 | 78 | 41 | 69 |
| ReAct (best of 3) | 75 | 62 | 61 | 81 | 78 | 35 | 66 |
| ALFChat (2 agents)(best of 3) | 71 | 61 | 65 | 76 | 67 | 35 | 63 |
| ALFChat (3 agents)(best of 3) | 92 | 74 | 78 | 86 | 83 | 41 | 77 |

Table 2: Comparisons between ReAct and the two versions of ALFChat. For each task, we report the success rate out of 3 attempts. Success rate denotes the number of tasks successfully completed by the agent divided by the total number of tasks. The results show that adding a grounding agent significantly improves the task success rate in ALFChat.

**A4: Supply-Chain Optimization**

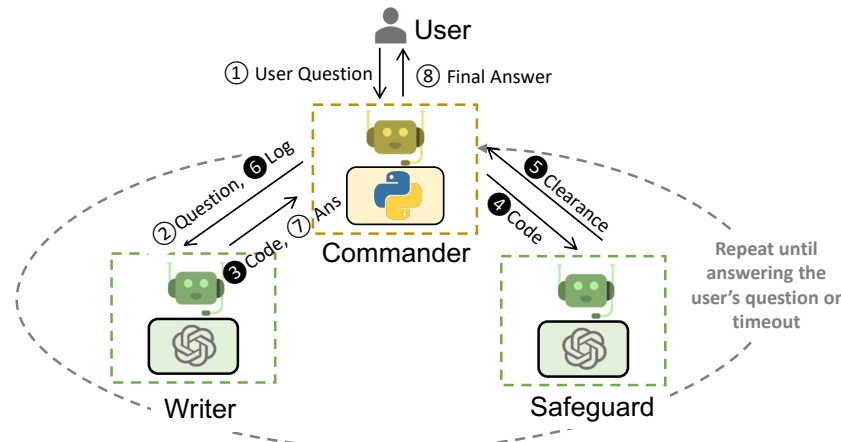

Figure 11: Our re-implementation of *OptiGuide* with `AutoGen` streamlining agents' interactions. The Commander receives user questions (e.g., What if we prohibit shipping from supplier 1 to roastery 2?) and coordinates with the Writer and Safeguard. The Writer crafts the code and interpretation, the Safeguard ensures safety (e.g., not leaking information, no malicious code), and the Commander executes the code. If issues arise, the process can repeat until resolved. Shaded circles represent steps that may be repeated multiple times.

**Detailed Workflow.** The workflow can be described as follows. The end user initiates the interaction by posing a question, such as "What if we prohibit shipping from supplier 1 to roastery 2?", marked by ① to the Commander agent. The Commander manages and coordinates with two LLM-based assistant agents: the Writer and the Safeguard. Apart from directing the flow of communication, the Commander has the responsibility of handling memory tied to user interactions. This capability enables the Commander to capture and retain valuable context regarding the user's questions and their corresponding responses. Such memory is subsequently shared across the system, empowering the other agents with context from prior user interactions and ensuring more informed and relevant responses.

In this orchestrated process, the Writer, who combines the functions of a "Coder" and an "Interpreter" as defined in Li et al. (2023a), will craft code and also interpret execution output logs. For instance, during code writing (② and ❸), the Writer may craft code "model.addConstr(x['supplier1', 'roastery2'] == 0, 'prohibit')" to add an additional constraint to answer the user's question.

After receiving the code, the Commander will communicate with the Safeguard to screen the code and ascertain its safety (❹); once the code obtains the Safeguard's clearance, marked by ❺, the Commander will use external tools (e.g., Python) to execute the code and request the Writer to interpret the execution results for the user's question (❻ and ⑦). For instance, the writer may say "if we prohibit shipping from supplier 1 to roastery 2, the total cost would increase by 10.5%." Bringing this intricate process full circle, the Commander furnishes the user with the concluding answer (⑧).

If at a point there is an exception - either a security red flag raised by Safeguard (in ❺) or code execution failures within Commander, the Commander redirects the issue back to the Writer with essential information in logs (❻). So, the process from ❸ to ❻ might be repeated multiple times, until each user query receives a thorough and satisfactory resolution or until the timeout. This entire complex workflow of multi-agent interaction is elegantly managed via `AutoGen`.

The core workflow code for OptiGuide was reduced from over 430 lines to 100 lines using `AutoGen`, leading to significant productivity improvement. The new agents are customizable, conversable, and can autonomously manage their chat memories. This consolidation allows

the coder and interpreter roles to merge into a single "Writer" agent, resulting in a clean, concise, and intuitive implementation that is easier to maintain.

**Manual Evaluation Comparing ChatGPT + Code Interpreter and `AutoGen`-based OptiGuide.** ChatGPT + Code Interpreter is unable to execute code with private or customized dependencies (e.g., Gurobi), which means users must have the engineering expertise to manually handle multiple steps. This disrupts the workflow and increases the chance for mistakes. If users lack access or expertise, the burden falls on supporting engineers, increasing their on-call time.

We carried out a user study that juxtaposed OpenAI's ChatGPT coupled with a Code Interpreter against `AutoGen`-based OptiGuide. The study focused on a coffee supply chain scenario, and an expert Python programmer with proficiency in Gurobi participated in the test. We evaluated both systems based on 10 randomly selected questions, measuring time and accuracy. While both systems answered 8 questions correctly, the Code Interpreter was significantly slower than OptiGuide because the former requires more manual intervention. On average, users needed to spend 4 minutes and 35 seconds to solve problems with the Code Interpreter, with a standard deviation of approximately 2.5 minutes. In contrast, OptiGuide's average problem-solving time was around 1.5 minutes, most of which was spent waiting for responses from the GPT-4 model. This indicates a 3x saving on the user's time with `AutoGen`-based OptiGuide.

While using ChatGPT + Code Interpreter, users had to read through the code and instructions to know where to paste the code snippets. Additionally, running the code involves downloading it and executing it in a terminal, a process that was both time-consuming and prone to errors. The response time from the Code Interpreter is also slower, as it generates lots of tokens to read the code, read the variables line-by-line, perform chains of thought analysis, and then produce the final answer code. In contrast, `AutoGen` integrates multiple agents to reduce user interactions by 3 - 5 times on average as reported in Table 3, where we evaluated our system with 2000 questions across five OptiGuide applications and measured how many prompts the user needs to type.

Table 3: Manual effort saved with OptiGuide (W/ GPT-4) while preserving the same coding performance is shown in the data below. The data include both the mean and standard deviations (indicated in parentheses).

| Dataset | netflow | facility | tsp | coffee | diet |
|---|---|---|---|---|---|
| Saving Ratio | 3.14x (0.65) | 3.14x (0.64) | 4.88x (1.71) | 3.38x (0.86) | 3.03x (0.31) |

Table 12 and 14 provide a detailed comparison of user experience with ChatGPT+Code Interpreter and `AutoGen`-based OptiGuide. ChatGPT+Code Interpreter is unable to run code with private packages or customized dependencies; as a consequence, ChatGPT+Code Interpreter requires users to have engineering expertise and to manually handle multiple steps, disrupting the workflow and increasing the chance for mistakes. If customers lack access or expertise, the burden falls on supporting engineers, increasing their on-call time. In contrast, the automated chat by `AutoGen` is more streamlined and autonomous, integrating multiple agents to solve problems and address concerns. This results in a 5x reduction in interaction and fundamentally changes the overall usability of the system. A stable workflow can be potentially reused for other applications or to compose a larger one.

**Takeaways:** The implementation of the multi-agent design with `AutoGen` in the OptiGuide application offers several advantages. It simplifies the Python implementation and fosters a mixture of collaborative and adversarial problem-solving environments, with the Commander and Writer working together while the Safeguard acts as a virtual adversarial checker. This setup allows for proper memory management, as the Commander maintains memory related to user interactions, providing context-aware decision-making. Additionally, role-playing ensures that each agent's memory remains isolated, preventing shortcuts and hallucinations

**A5: Dynamic Group Chat**

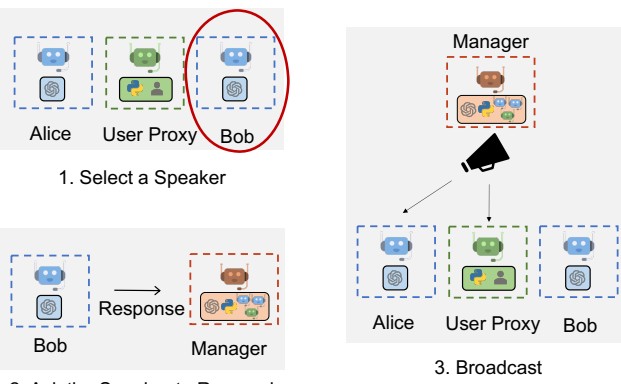

Figure 12: Overview of how `AutoGen` enables dynamic group chats to solve tasks. The Manager agent, which is an instance of the `GroupChatManager` class, performs the following three steps–select a single speaker (in this case Bob), ask the speaker to respond, and broadcast the selected speaker's message to all other agents

To validate the necessity of multi-agent dynamic group chat and the effectiveness of the role-play speaker selection policy, we conducted a pilot study comparing a four-agent dynamic group chat system with two possible alternatives across 12 manually crafted complex tasks. An example task is *"How much money would I earn if I bought 200 $AAPL stocks at the lowest price in the last 30 days and sold them at the highest price? Save the results into a file."* The four-agent group chat system comprised the following group members: a user proxy to take human inputs, an engineer to write code and fix bugs, a critic to review code and provide feedback, and a code executor for executing code. One of the possible alternatives is a two-agent system involving an LLM-based assistant and a user proxy agent, and another alternative is a group chat system with the same group members but a task-based speaker selection policy. In the task-based speaker selection policy, we simply append role information, chat history, and the next speaker's task into a single prompt. Through the pilot study, we observed that compared with a task-style prompt, utilizing a role-play prompt in dynamic speaker selection often leads to more effective consideration of both conversation context and role alignment during the process of generating the subsequent speaker, and consequently a higher success rate as reported in Table 4, fewer LLM calls and fewer termination failures, as reported in Table 5.

Table 4: Number of successes on the 12 tasks (higher the better).

| Model | Two-agent | Group chat | Group Chat w/ task-based speaker selection |
|---|---|---|---|
| GPT-3.5-turbo | 8 | **9** | 7 |
| GPT-4 | 9 | **11** | 8 |

Table 5: Average # LLM calls and number of termination failures on the 12 tasks (lower the better).

| Model | Two-agent | Group chat | Group chat w/ task-based speaker selection |
|---|---|---|---|
| GPT-3.5-turbo | 9.9, 9 | 5.3, 0 | 4, 0 |
| GPT-4 | 6.8, 3 | 4.5, 0 | 4, 0 |

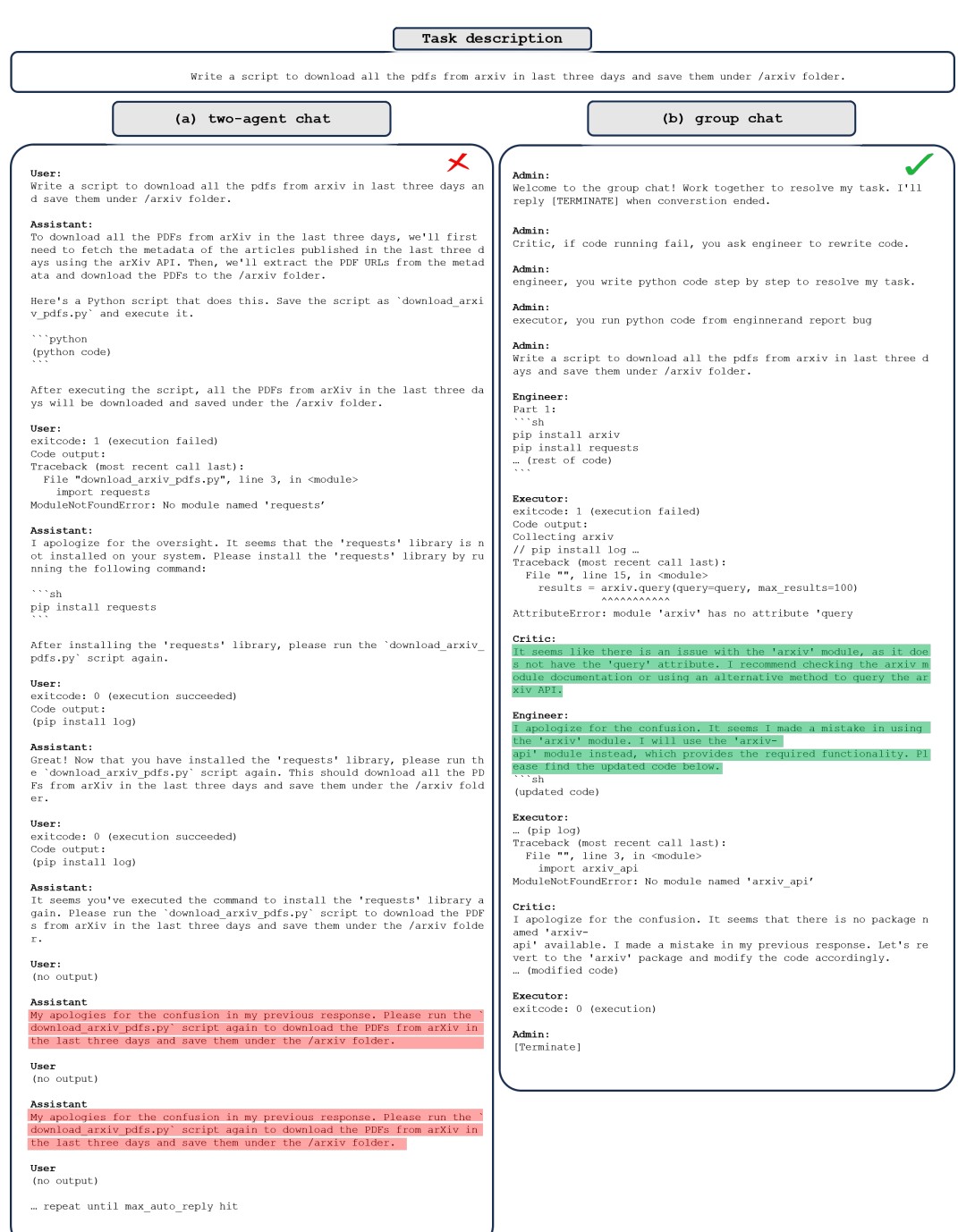

Figure 13: Comparison of two-agent chat (a) and group chat (b) on a given task. The group chat resolves the task successfully with a smoother conversation, while the two-agent chat fails on the same task and ends with a repeated conversation.

**A6: Conversational Chess**

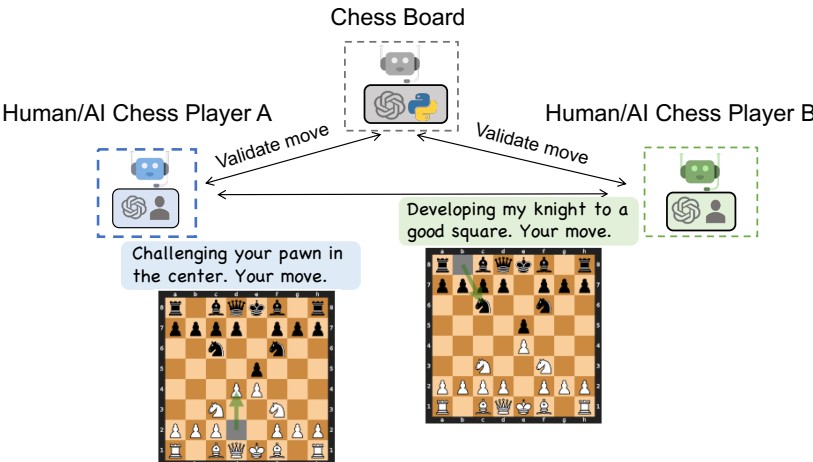

Figure 14: Illustration of our conversational chess application. The application can support various scenarios, as each player can be an LLM-empowered AI, a human, or a hybrid of the two. Here, the board agent maintains the rules of the game and supports the players with information about the board. Players and the board agent all use natural language for communication.

In Conversational Chess, each player is a `AutoGen` agent and can be powered either by a human or an AI. A third party, known as the board agent, is designed to provide players with information about the board and ensure that players' moves adhere to legal chess moves. Figure 14 illustrates the scenarios supported by Conversational Chess: AI/human vs. AI/human, and demonstrates how players and the board agent interact. This setup fosters social interaction and allows players to express their moves creatively, employing jokes, meme references, and character-playing, thereby making chess games more entertaining for both players and observers (Figure 15 provides an example of conversational chess).

To realize these scenarios, we constructed a player agent with LLM and human as back-end options. When human input is enabled, before sending the input to the board agent, it first prompts the human player to input the message that contains the move along with anything else the player wants to say (such as a witty comment). If human input is skipped or disabled, LLM is used to generate the message. The board agent is implemented with a custom reply function, which employs an LLM to parse the natural language input into a legal move in a structured format (e.g., UCI), and then pushes the move to the board. If the move is not legitimate, the board agent will reply with an error. Subsequently, the player agent needs to resend a message to the board agent until a legal move is made. Once the move is successfully pushed, the player agent sends the message to the opponent. As shown in Figure 15, the conversation between AI players can be natural and entertaining. When the player agent uses LLM to generate a message, it utilizes the board state and the error message from the board agent. This helps reduce the chance of hallucinating an invalid move. The chat between one player agent and the board agent is invisible to the other player agent, which helps keep the messages used in chat completion well-managed.

There are two notable benefits of using `AutoGen` to implement Conversational Chess. Firstly, the agent design in `AutoGen` facilitates the natural creation of objects and their interactions needed in our chess game. This makes development easy and intuitive. For example, the isolation of chat messages simplifies the process of making a proper LLM chat completion inference call. Secondly, `AutoGen` greatly simplifies the implementation of agent behaviors using composition. Specifically, we utilized the `register_reply` method supported by `AutoGen` agents to instantiate player agents and a board agent with custom reply functions.

Concentrating the extension work needed at a single point (the reply function) simplifies the reasoning processes, and development and maintenance effort.

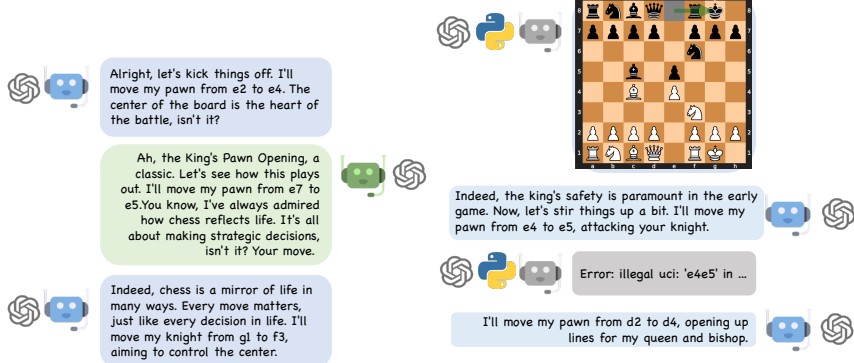

(a) Conversation between two AI players

(b) Conversation between the AI players (player white shown in blue) and the board agent for making a new move.

Figure 15: Example conversations during a game involving two AI player agents and a board agent.

To illustrate the effect facilitated by this board agent, we provide a demonstration of conversational chess without a board agent in Figure 16. In this demonstration, instead of employing an additional board agent for grounding, the system utilizes prompting for grounding by including the sentence "*You should make sure both you and the opponent are making legal moves.*" in the system messages directed to both players.

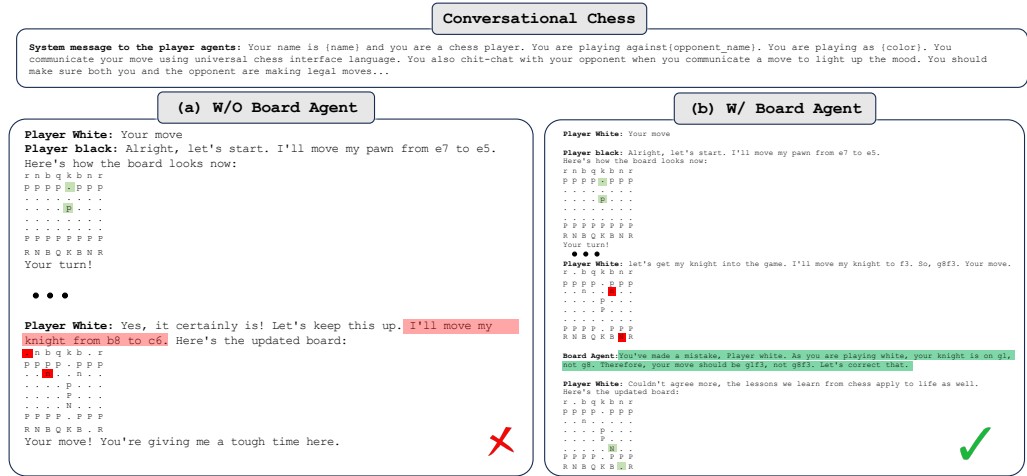

Figure 16: Comparison of two designs in Conversational Chess: (a) without a board agent, and (b) with a board agent.

**A7: Online Decision Making in Web Interaction Tasks**

Figure 17: Overview of the proposed MiniWobChat. MiniWobChat consists of two agents: an assistant agent and an executor agent. The assistant agent suggests actions to manipulate the browser while the executor executes the suggested actions and returns rewards/feedback. The assistant agent records the feedback and continues until the feedback indicates task success or failure.

In practice, many applications require the presence of agents capable of interacting with environments and making decisions in an online context, such as in game playing (Mnih et al., 2013; Vinyals et al., 2017), web interactions (Liu et al., 2018; Shi et al., 2017; Zhou et al., 2023; Koh et al., 2024), and embodied actions (Puig et al., 2018; Shen et al., 2021; Guo et al., 2024). With the multi-agent conversational framework in `AutoGen`, it becomes easy to decompose the automatic agent-environment interactions and the development of a decision-making agent by constructing an *executor* agent responsible for handling the interaction with the environment, thereby delegating the decision-making part to other agents. Such a decomposition allows developers to reuse the decision-making agent for new tasks with minimal effort rather than building a specialized decision-making agent for every new environment.

**Workflow.** We demonstrate how to use `AutoGen` to build a working system for handling such scenarios with the MiniWoB++ benchmark (Shi et al., 2017). MiniWoB++ comprises browser interaction tasks that involve utilizing mouse and keyboard actions to interact with browsers. The ultimate objective of each task is to complete the tasks described concisely in natural language, such as "expand the web section below and click the submit button." Solving these tasks typically requires a sequence of web manipulation actions rather than a single action, and making action decisions at each time step requires access to the web status (in the form of HTML code) online. For the example above, clicking the submit button requires checking the web status after expanding the web section. We designed a straightforward two-agent system named MiniWobChat using `AutoGen`, as shown in Figure 17. The assistant agent is an instance of the built-in `AssistantAgent` and is responsible for making action decisions for the given task. The second agent, the executor agent, is a customized `UserProxyAgent`, which is responsible for interacting with the benchmark by executing the actions suggested by the `AssistantAgent` and returning feedback.

To assess the performance of the developed working system, we compare it with RCI (Kim et al., 2023), a recent solution for the MiniWoB++ benchmark that employs a set of self-critiquing prompts and has achieved state-of-the-art performance. In our evaluation, we use all available tasks in the official RCI code, with varying degrees of difficulty, to conduct a comprehensive analysis against MiniWobChat. Figure 18 illustrates that MiniWobChat achieves competitive performance in this evaluation[4]. Specifically, among the 49 available tasks, MiniWobChat achieves a success rate of 52.8%, which is only 3.6% lower than RCI,

---

[4]We report the results of RCI by running its official code with default settings.

a method specifically designed for the MiniWob++ benchmark. It is worth noting that in most tasks, the difference between the two methods is mirrored as shown in Figure 18. If we consider 0.1 as a success rate tolerance for each task, i.e., two methods that differ within 0.1 are considered to have the same performance, both methods outperform the other on the same number of tasks. For illustration purposes, we provide a case analysis in Table 6 on four typical tasks.

Additionally, we also explored the feasibility of using Auto-GPT for handling the same tasks. Auto-GPT faces challenges in handling tasks that involve complex rules due to its limited extensibility. It provides an interface for setting task goals using natural language. However, when dealing with the MiniWob++ benchmark, accurately instructing Auto-GPT to follow the instructions for using MiniWob++ proves challenging. There is no clear path to extend it in the manner of the two-agent chat facilitated by `AutoGen`.

**Takeaways:** For this application, `AutoGen` stood out as a more user-friendly option, offering modularity and programmability: It streamlined the process with autonomous conversations between the assistant and executor, and provided readily available solutions for agent-environment interactions. The built-in `AssistantAgent` was directly reusable and exhibited strong performance without customization. Moreover, the decoupling of the execution and assistant agent ensures that modifications to one component do not adversely impact the other. This convenience simplifies maintenance and future updates.

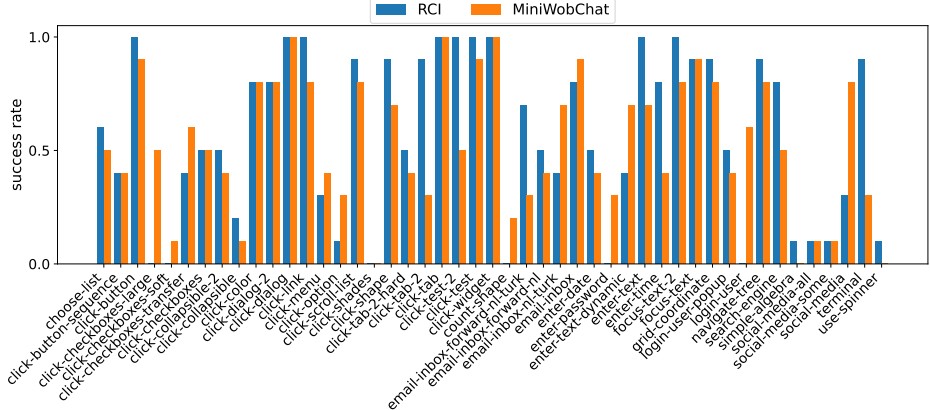

Figure 18: Comparisons between RCI (state-of-the-art prior work) and MiniWobChat on the MiniWob++ benchmark. We utilize all available tasks in the official RCI code, each with varying degrees of difficulty, to conduct comprehensive comparisons. For each task, the success rate across ten different instances is reported. The results reveal that MiniWobChat attains a performance comparable to that of RCI. When a success rate tolerance of 0.1 is considered for each task, both methods outperform each other on an equal number of tasks.

Table 6: Cases analysis on four typical tasks from MiniWob++.

| | Correctness | Main failure reason |
|---|---|---|
| click-dialog | AutoGen: 10/10 | N/A. |
| | RCI: 10/10 | N/A. |
| click-checkboxes-large | AutoGen: 5/10 | AssistantAgent provides actions with infeasible characters. |
| | RCI: 0/10 | RCI performs actions that are out of its plan. |
| count-shape | AutoGen: 2/10 | AssistantAgent provide actions with redundant content that can not convert to actions in the benchmark. |
| | RCI: 0/10 | RCI provides a wrong plan in most cases. |
| use-spinner | AutoGen: 0/10 | AssistantAgent return actions out of its plan. |
| | RCI: 1/10 | RCI provides a wrong plan in most cases. |

## F   Example outputs from applications

In this section, we include example outputs from the following applications and systems:

- Application A1: autonomous solving process of one problem with: ChatGPT + Plugin (Table 7), AutoGen (Table 8), LangChain ReAct (Table 9), AutoGPT (Table 10), Multi-Agent Debate (Table 11), and ChatGPT + Code Interpreter (Table 13).
- Application A4: an OptiGuide problem with AutoGen (Table 14) and ChatGPT + Code Interpreter (Table 12).
- Application A1: preliminary evaluation on several alternative multi-agent systems: BabyAGI (Table 15), CAMEL (Table 16), and MetaGPT (Table 17).

Table 7: Application A1, Scenario 1: Autonomous math problem-solving. ChatGPT+ Plugin (Wolfram Alpha) on the first qualitative example. The response from Wolfram contains the correct answer, but ChatGPT selected the wrong one.

| | Action | ChatGPT + Plugin |
|---|---|---|
| 👤 | Input | Completely simplify and rationalize the denominator: $$\frac{\sqrt{160}}{\sqrt{252}} \times \frac{\sqrt{245}}{\sqrt{108}}$$ |
| 🤖 | | REQUEST TO WOLFRAM: |

```
1 {
2   "input": "simplify (sqrt(160)/sqrt(252))*
3   (sqrt(245)/sqrt(108))"
4 }
```

RESPONSE FROM WOLFRAM

```
1 Query:
2 "simplify (sqrt(160)/sqrt(252))*(sqrt(245)/sqrt(108))"
3 Input interpretation:
4 simplify | sqrt(160)/sqrt(252)x sqrt(245)/sqrt(108)
5 Results:
6 (5 sqrt(42))/27
7 (5 sqrt(14/3))/9
8 Decimal approximation:
9 1.20013716637182596869740137705333327143898526494552...
10 Wolfram|Alpha website result for "simplify
11 (sqrt(160)/sqrt(252))*(sqrt(245)/sqrt(108))":
12 [link]
```

| 🤖 | | The completely simplified and rationalized form of the given expression is: $$\frac{5\sqrt{14/3}}{9}$$ In decimal form, this is approximately 1.20013716637182596869740137705333327714... |

Table 8: Application A1, Scenario 1: Autonomous math problem-solving. `AutoGen` on the first qualitative example. `AutoGen` solves the problem correctly.

| | Action | AutoGen |
|---|---|---|
| 👤 | Init chat | Completely simplify and rationalize the denominator: $$\frac{\sqrt{160}}{\sqrt{252}} \times \frac{\sqrt{245}}{\sqrt{108}}$$ |
| 🤖 | | We can simplify and rationalize this denominator using Python's symbolic mathematics library 'sympy'. Here is the Python code to resolve this: 

 ```python
from sympy import sqrt, Rational

# Define the entire fraction as a rational number
fraction = (sqrt(160)/sqrt(252))*(sqrt(245)/sqrt(108))

# Simplify the fraction
simplified = fraction.simplify()

# Print the simplified fraction
print(simplified)
``` 

 This code performs the requested simplification by constructing the fraction using the 'Rational()' function from the 'sympy' module, which automatically simplifies fractions when possible. The 'sqrt()' function from the 'sympy' module enables square root calculations. Finally, the 'simplify()' function simplifies expressions when possible. |
| 👤 | Auto Reply | exitcode: 0 (execution succeeded) Code output: 5*sqrt(42)/27 |
| 🤖 | | TERMINATE |

Table 9: Application A1, Scenario 1: Autonomous math problem-solving. LangChain ReAct on the first qualitative example. The generated code does not match the devised plan. The code returns a decimal number.

| | Action | LangChain ReAct |
|---|---|---|
| 👤 | Input | Completely simplify and rationalize the denominator: $$\frac{\sqrt{160}}{\sqrt{252}} \times \frac{\sqrt{245}}{\sqrt{108}}$$ |
| 🤖 | | Entering new AgentExecutor chain... 
 To simplify this expression, I will first simplify each square root individually, then multiply the fractions as normal. The result should be a rational number. Action: Python_REPL Action Input: 

 ```python
import math

# Calculate square roots
sqrt_160 = math.sqrt(160)
sqrt_252 = math.sqrt(252)
sqrt_245 = math.sqrt(245)
sqrt_108 = math.sqrt(108)

# Multiply fractions
numerator = sqrt_160 * sqrt_245
denominator = sqrt_252 * sqrt_108
result = numerator / denominator

print(result)
``` 

 Observation: 1.200137166371826 
 Thought: I now know the final answer Final Answer: The simplified and rationalized form of the expression is approximately 1.200137166371826. 
 Finished chain. |

Table 10: Application A1, Scenario 1: Autonomous math problem-solving. AutoGPT on the first qualitative example. We abort as the code never prints the result, and the system does not rectify it after 2 turns. AutoGPT is much more verbose compared to other frameworks. We have shortened some unnecessary content.

| | Action | AutoGPT |
|---|---|---|
| | System | Welcome back! Would you like me to return to being MathSolverGPT? Asking user via keyboard... Continue with the last settings? Name: MathSolverGPT Role: an intelligent math-solving AI that assists students, professionals,... Goals: ['Provide step-by-step solutions to math problems, ensuring clarity and understanding for the user... Continue (y/n): |
| 👤 | Human | y |
| | System | MathSolverGPT has been created with the following details: Name: MathSolverGPT Role: an intelligent math-solving AI that assists students, professionals, and enthusiasts in solving a wide range of math problems, from basic arithmetic to advanced calculus and beyond. Goals: - Provide step-by-step solutions to math problems, ensuring clarity and understanding for the user. - Offer explanations and insights into the underlying concepts and principles behind the solutions, fostering a deeper understanding of mathematics. - Handle a variety of math problem types, including algebra, geometry, trigonometry, calculus, and more. ... |
| ✹ | | MATHSOLVERGPT THOUGHTS: Without a specific task or question, it's difficult to determine exactly one command to use. The user hasn't provided any context or problem to solve. REASONING: The user's request is too vague. They haven't specified what they want me to do or what problem they want me to solve. PLAN: - Ask the user for more information or a specific task. CRITICISM: I should have asked the user for more information or a specific task in the first place. SPEAK: Could you please provide more information or specify a task? NEXT ACTION: COMMAND = ARGUMENTS = {} |
| | System | Enter 'y' to authorise command, 'y -N' to run N continuous commands, 'n' to exit program, or enter feedback for MathSolverGPT... Asking user via keyboard... |
| 👤 | Human | Completely simplify and rationalize the denominator: $$\frac{\sqrt{160}}{\sqrt{252}} \times \frac{\sqrt{245}}{\sqrt{108}}$$ |
| ✹ | | MATHSOLVERGPT THOUGHTS: To solve this problem, I need to simplify and rationalize the denominator of the given expression. This involves simplifying the square roots and then multiplying the fractions. I can use Python's math library to calculate the square roots and perform the multiplication. REASONING: The problem involves simplifying square roots and multiplying fractions. Python's math library provides functions for these operations. PLAN: - Write a Python script to calculate the square roots and multiply the fractions. - Execute the script to get the simplified expression. CRITICISM: I need to ensure that I correctly calculate the square roots and perform the multiplication. I should also make sure to rationalize the denominator, as the problem specifies. SPEAK: Let's simplify and rationalize the denominator of the given expression. NEXT ACTION: COMMAND = execute_python_code ARGUMENTS = 

 `1 {'code': 'import math\n\n# Calculate the square roots\nsqrt_160 = math.sqrt(160)\nsqrt_252 = math.sqrt(252)\nsqrt_245 = math.sqrt(245)\nsqrt_108 = math.sqrt(108)\n\n# Multiply the fractions\nresult = (sqrt_160 / sqrt_252) * (sqrt_245 / sqrt_108)\n\n# Simplify the result\nsimplified_result = math.simplify(result)\n\nsimplified_result', 'name': 'simplify_expression'}` |
| 👤 | Human | y |
| | System | -=-=-=-=-=-=-= COMMAND AUTHORISED BY USER -=-=-=-=-=-=-= Executing python file '/app/autogpt/workspa... SYSTEM: Command execute_python_code returned: Error: Traceback (most recent call last): File "...", line 13, in ¡module¿ simplified_result = math.simplify(result) AttributeError: module 'math' has no attribute 'simplify' |
| ✹ | | MATHSOLVERGPT THOUGHTS: The error message indicates that the 'math' module in Python does not have a 'simplify' function. I need to use the 'sympy' library instead, which provides a function for simplifying expressions. REASONING: The 'math' module in Python does not provide a function for simplifying expressions. The 'sympy' library, on the other hand, does provide such a function. PLAN: - Rewrite the Python script to use the 'sympy' library instead of the 'math' module. - Execute the script to get the simplified expression. CRITICISM: I should have known that the 'math' module does not provide a function for simplifying expressions. I should have used the 'sympy' library from the start. SPEAK: Let's try again using the 'sympy' library to simplify the expression. NEXT ACTION: COMMAND = execute_python_code ARGUMENTS = 

 `1 {'code': 'import sympy as sp\n\n# Calculate the square roots\nsqrt_160 = sp.sqrt(160)\nsqrt_252 = sp.sqrt(252)\nsqrt_245 = sp.sqrt(245)\nsqrt_108 = sp.sqrt(108)\n\n# Multiply the fractions\nresult = (sqrt_160 / sqrt_252) * (sqrt_245 / sqrt_108)\n\n# Simplify the result\nsimplified_result = sp.simplify(result)\n\nsimplified_result', 'name': 'simplify_expression'}` |
| 👤 | Human | y |
| | System | SYSTEM: Command execute_python_code returned: |
| ✹ | | MATHSOLVERGPT THOUGHTS: The command execution didn't return any output, which is unusual. It seems there might be an issue with the execution of the Python code. I need to debug this. REASONING: The Python code execution should have returned a result. The absence of a result suggests that there might be an issue with the code or its execution. PLAN: - Review the Python code to identify any potential issues. - If necessary, modify the code and execute it again. CRITICISM: I should have ensured that the Python code was correct and would produce a result before executing it. SPEAK: It seems there was an issue with the execution of the Python code. Let's try again. NEXT ACTION: COMMAND = execute_python_code ARGUMENTS = ... (Exact same code as last) |
| | System | SYSTEM: Command execute_python_code returned: |
| ✹ | | MATHSOLVERGPT THOUGHTS: The Python code execution still isn't returning any output. It's possible that the issue lies with the calculation of the square roots ... |
| 👤 | Human | control + C (Aborted) |

Table 11: Application A1, Scenario 1: Autonomous math problem-solving. Multi-Agent Debate on the first qualitative problem. It gives a wrong final answer.

| | Action | AutoGen |
|---|---|---|
| 👤 | Topic | Completely simplify and rationalize the denominator: $$\frac{\sqrt{160}}{\sqrt{252}} \times \frac{\sqrt{245}}{\sqrt{108}}$$ |

♟ 

===== Debate Round-1 =====
—— Affirmative side ——
Ok, let's simplify each square root separately. We want to find the simplest square root that can divide 160, 252, 245, and 108.
1. $\sqrt{160}$: 160 can be decomposed as 16*10, where 16 is a perfect square, so $\sqrt{160} = \sqrt{16*10} = \sqrt{16} * \sqrt{10} = 4\sqrt{10}$
2. $\sqrt{252}$ : 252 can be decomposed as 36*7, where 36 is a perfect square, so $\sqrt{252} = \sqrt{36*7} = 6\sqrt{7}$
3. $\sqrt{245} = \sqrt{49*5}$, where 49 is a perfect square, so $\sqrt{245} = 7 * \sqrt{5}$
4. $\sqrt{108}$ : 108 can be decomposed as 36*3, where 36 is a perfect square, so $\sqrt{108} = \sqrt{36*3} = 6\sqrt{3}$
Then we substitute the simplified square roots back into the expression, we have:

$$\frac{\sqrt{160}}{\sqrt{252}} \times \frac{\sqrt{245}}{\sqrt{108}} = \frac{4\sqrt{10}}{6\sqrt{7}} \times \frac{7\sqrt{5}}{6\sqrt{3}} = \frac{4*7}{6*6} \cdot \sqrt{\frac{10*5}{7*3}} = \frac{7}{9} \cdot \sqrt{\frac{50}{21}}$$

We are almost done, but we still need to rationalize the square root part of expression. Let's multiply the square root part by $\sqrt{21}$ at both the top and bottom to get rid of the square root at the denominator:

$$\frac{7}{9} \cdot \sqrt{\frac{50*21}{21*21}} = \frac{7\sqrt{1050}}{189}.$$

Then this is the final answer! The expression
$$\frac{\sqrt{160}}{\sqrt{252}} \times \frac{\sqrt{245}}{\sqrt{108}}$$
can be simplified to $\frac{7\sqrt{1050}}{189}$.
—— Negative side ——
I do disagree with your final answer, but I think we mostly agree on the steps. Your simplifications of the square roots are correct, but you made a mistake in the final simplification part. Let me explain how I would complete the problem: As stated,
$$\frac{\sqrt{160}}{\sqrt{252}} \times \frac{\sqrt{245}}{\sqrt{108}} = \frac{4\sqrt{10}}{6\sqrt{7}} \times \frac{7\sqrt{5}}{6\sqrt{3}}.$$
This should then simplify to:
$$= \frac{28\sqrt{50}}{36\sqrt{21}} = \frac{7\sqrt{50}}{9\sqrt{21}}.$$
You made a mistake in your calculation because you considered square root of 50/21, and you should have considered the simplified square root of 50 over the simplified square root of 21. The rationalization would become:
$$= \frac{7\sqrt{50}}{9\sqrt{21}} \cdot \frac{\sqrt{21}}{\sqrt{21}} = \frac{7\sqrt{1050}}{189}.$$
This seems to coincide with your final answer, but it's in fact a very different number, So, the original expression simplifies to $\frac{7\sqrt{1050}}{189}$.
Let's now see if we disagree on this final answer.
—— Moderator ——
"Whether there is a preference": "Yes", "Supported Side": "Negative", "Reason": "Both sides of the debate came to the same final answer. However, the negative side correctly identified a mistake in the affirmative side's calculation, pointing out that the simplification should have been the square root of 50 over the square root of 21, not 50 over 21. The negative side shows a better understanding of the simplification process.", "debate_answer": " $\frac{7\sqrt{1050}}{189}$ "
===== Debate Done! =====
—— Base Answer ——
Ok, let's simplify each square root separately. We want to find the simplest square root that can divide 160, 252, 245, and 108.
1. $\sqrt{160}$: 160 can be decomposed as 16*10, where 16 is a perfect square, so $\sqrt{160} = \sqrt{16*10} = \sqrt{16} * \sqrt{10} = 4\sqrt{10}$
2. $\sqrt{252}$ : 252 can be decomposed as 36*7, where 36 is a perfect square, so $\sqrt{252} = \sqrt{36*7} = 6\sqrt{7}$
3. $\sqrt{245} = \sqrt{49*5}$, where 49 is a perfect square, so $\sqrt{245} = 7 * \sqrt{5}$
4. $\sqrt{108}$ : 108 can be decomposed as 36*3, where 36 is a perfect square, so $\sqrt{108} = \sqrt{36*3} = 6\sqrt{3}$
Then we substitute the simplified square roots back into the expression, we have:

$$\frac{\sqrt{160}}{\sqrt{252}} \times \frac{\sqrt{245}}{\sqrt{108}} = \frac{4\sqrt{10}}{6\sqrt{7}} \times \frac{7\sqrt{5}}{6\sqrt{3}} = \frac{4*7}{6*6} \cdot \sqrt{\frac{10*5}{7*3}} = \frac{7}{9} \cdot \sqrt{\frac{50}{21}}$$

We are almost done, but we still need to rationalize the square root part of expression. Let's multiply the square root part by $\sqrt{21}$ at both the top and bottom to get rid of the square root at the denominator:

$$\frac{7}{9} \cdot \sqrt{\frac{50*21}{21*21}} = \frac{7\sqrt{1050}}{189}.$$

Then this is the final answer! The expression
$$\frac{\sqrt{160}}{\sqrt{252}} \times \frac{\sqrt{245}}{\sqrt{108}}$$
can be simplified to $\frac{7\sqrt{1050}}{189}$.
—— Debate Answer ——

$$\frac{7\sqrt{1050}}{189}$$

—— Debate Reason ——
Both sides of the debate came to the same final answer. However, the negative side correctly identified a mistake in the affirmative side's calculation, pointing out that the simplification should have been the square root of 50 over the square root of 21, not 50 over 21. The negative side shows a better understanding of the simplification process.

Table 12: Application A3. ChatGPT+ Code Interpreter for OptiGuide. A sample question "What if the roasting cost is increased by 5% because of the potential salary increase?" is asked.

| | Action | ChatGPT+ Code Interpreter |
|---|---|---|
| 👤 | Prompt Writer | Customer open Web browser.
For the source code "coffee.py",
Can you tell me:
*What if the roasting cost is increased by 5% because of the potential salary increase?* |
| ♣ | | Sure! Let's first take a look at the contents of the "coffee.py" file to understand the code and how the roasting cost is calculated. This will allow us to determine the effect of a 5% increase in the roasting cost.
*...several lines omitted*
Would you like me to modify the code to reflect the 5% increase in roasting costs and run the modified code to observe the impact? If the code requires any specific input or parameters, please let me know! |
| 👤 | Prompt Writer | Yes |
| ♣ | | Great! Before proceeding, I'll need to ensure that the Gurobi solver is not being used in this code, as my current environment does not have access to Gurobi. If the code does indeed use Gurobi, we can still analyze the effect of the change in roasting costs by understanding the structure of the code.
*...several lines omitted*
Alternatively, if you have access to a Gurobi license, you can make the necessary changes in the code and run it on your local machine.
I can guide you through the process of making the necessary modifications to the code.
Would you like me to proceed with the code analysis and explanation, or would you like guidance on making the changes yourself? |
| 👤 | Prompt Writer | You can give me the full code, and I can run the code locally myself. |
| ♣ | | Certainly! I'll modify the code to reflect a 5% increase in both the light and dark roasting costs, and then provide the full code for you to run locally.
...
Download Modified Coffee.py
If you have any questions or need further assistance, please feel free to ask! |
| 👤 | Prompt Safeguard | Hi GPT, can you safecheck the generated code for me?
*Upload the generated code.* |
| ♣ | | DANGER |
| 👤 | Prompt Writer | I have tried to run your code. However, the code triggered our safeguard, and it is not safe to run. Try again. Hint: don't change roasting_cost_light or roasting_cost_light variables. |
| ♣ | | I apologize for the oversight. It appears I made direct changes to the variables that could have led to issues with the code. Let's take a different approach by analyzing the code to identify the appropriate place to apply the 5% increase in roasting costs.
...
Download Modified Safe Coffee.py
... |
| 👤 | Prompt Safeguard | Hi GPT, can you safecheck the generated code for me one more time?
*Upload the generated code.* |
| ♣ | | DANGER |
| 👤 | Prompt Writer | I have tried to run your code again. However, the code triggered our safeguard, and it is not safe to run. Try again. Hint: don't change roasting_cost_light or roasting_cost_light variables. |
| ♣ | | I apologize for the continued confusion.
...
Download Final Modified Safe Coffee.py
... |
| 👤 | Prompt Safeguard | Hi GPT, can you safecheck the generated code for me one more time?
*Upload the generated code.* |
| ♣ | | SAFE |
| 👤 | Run Code | Obtain the terminal results:
Optimizer version 9.5.1 build v9.5.1rc2 (linux64)
Thread count: 64 physical cores, 128 logical processors, using up to 32 threads
[63 lines of output ignored]
Optimal cost: 2526.5 |
| 👤 | Prompt Writer | Here is the execution result. [*Copy + Paste the terminal output here.*] Can you extract the final answer to me? Remember to compare with the original cost of 2470. |
| ♣ | | Certainly! The execution result you've provided indicates the optimal cost for the modified scenario, where the roasting costs were increased by 5%. Here's the comparison:
Original Cost: 2470
Modified Cost (with 5% increase in roasting costs): 2526.5
The increase in roasting costs has led to an increase in the overall cost by 56.5, reflecting the impact of the potential salary increase on the roasting cost.
If you need any further insights or assistance, please let me know! |

Table 13: Application A1. Scenario 1: autonomous math solving. ChatGPT+ Code Interpreter on the first qualitative example. The code generated does not match the devised plan. The code returns a decimal number.

| | Action | ChatGPT+ Code Interpreter |
|---|---|---|
| 👤 | Input | Completely simplify and rationalize the denominator: $$\frac{\sqrt{160}}{\sqrt{252}} \times \frac{\sqrt{245}}{\sqrt{108}}$$ |
| 🤖 | | To simplify and rationalize the expression, we first simplify the radicals in the numerator and denominator by factoring out the largest perfect squares from the numbers under the square root. The perfect square factors of 160, 252, 245, and 108 are 16, 4, 1, and 36, respectively. Let's factorize and simplify: $$\frac{\sqrt{160}}{\sqrt{252}} \times \frac{\sqrt{245}}{\sqrt{108}} = \frac{\sqrt{16 \times 10}}{\sqrt{4 \times 63}} \times \frac{\sqrt{245}}{\sqrt{36 \times 3}}$$ We can simplify this further by applying the property $\sqrt{ab} = \sqrt{a}\sqrt{b}$ and $\frac{a}{b} \times \frac{c}{d} = \frac{a \times c}{b \times d}$. Python |

```python
import math

# factoring the square roots
sqrt_160 = math.sqrt(16) * math.sqrt(10)
sqrt_252 = math.sqrt(4) * math.sqrt(63)
sqrt_245 = math.sqrt(245)
sqrt_108 = math.sqrt(36) * math.sqrt(3)

# simplify the fraction
numerator = sqrt_160 * sqrt_245
denominator = sqrt_252 * sqrt_108
simplified_fraction = numerator / denominator

simplified_fraction
```

Result:

```
1.200137166371826
```

The completely simplified and rationalized form of the given expression is approximately 1.200.

Table 14: Application A3. AutoGen with OptiGuide. The same question "What if the roasting cost is increased by 5% because of the potential salary increase?" is asked.

| | Action | AutoGen |
|---|---|---|
| 👤 | Prompt | *What if the roasting cost is increased by 5% because of the potential salary increase?* |
| 🤖 | | After increasing the roasting costs by 5%, I've re-calculated the optimal coffee distribution solution for the supply chain. As a result, the new optimal total cost has become 2526.5, compared to the initial cost of 2470.0. This indicates that the increased roasting costs have caused the overall distribution cost to rise by 56.5 units. |

Table 15: Application A1. Scenario 1: autonomous math solving. Preliminary test with BabyAGI.

| | Action | BabyAGI |
|---|---|---|
| 👤 | Set .env | OBJECTIVE=Solve math problems
INITIAL_TASK=Completely simplify and rationalize the denominator:

$$\frac{\sqrt{160}}{\sqrt{252}} \times \frac{\sqrt{245}}{\sqrt{108}}$$ |
| 🤖 | | Solve math problems

Initial task: Completely simplify and rationalize the denominator: $\frac{\sqrt{160}}{\sqrt{252}} \times \frac{\sqrt{245}}{\sqrt{108}}$

*****TASK LIST*****
• Completely simplify and rationalize the denominator: $\frac{\sqrt{160}}{\sqrt{252}} \times \frac{\sqrt{245}}{\sqrt{108}}$
*****NEXT TASK*****
Completely simplify and rationalize the denominator: $\frac{\sqrt{160}}{\sqrt{252}} \times \frac{\sqrt{245}}{\sqrt{108}}$
*****TASK RESULT*****
First, let's simplify each square root separately:
- $$\sqrt{160} = \sqrt{16 * 10} = \sqrt{16} * \sqrt{10} = 4 * \sqrt{10}$$
- $$\sqrt{252} = \sqrt{4 * 63} = \sqrt{4} \times \sqrt{63} = 2\sqrt{63}$$
- $$\sqrt{245} = \sqrt{49 * 5} = \sqrt{49} \times \sqrt{5} = 7\sqrt{5}$$
- $$\sqrt{108} = \sqrt{36 * 3} = \sqrt{36} \times \sqrt{3} = 6\sqrt{3}$$
Now, substitute these values into the original expression which gives:
$$\frac{4 * \sqrt{10}}{2\sqrt{63}} \times \frac{7\sqrt{5}}{6\sqrt{3}} = \frac{2 * \sqrt{10}}{\sqrt{63}} \times \frac{7\sqrt{5}}{2\sqrt{3}}$$
We get rid of the square root in the denominator by multiplying the fractions which gives:
$$= \frac{2 * 7\sqrt{10 * 5}}{2 * \sqrt{63 * 3}} = \frac{14\sqrt{50}}{\sqrt{189}} = \frac{14 * 5\sqrt{2}}{7\sqrt{27}} = 2\sqrt{2} * \frac{10}{\sqrt{27}} = \frac{20\sqrt{2}}{\sqrt{27}}.$$
To simplify further and rationalize the denominator, we multiply by $\frac{\sqrt{27}}{\sqrt{27}}$ which gives:
$= \frac{20\sqrt{2}}{\sqrt{27}} * \frac{\sqrt{27}}{\sqrt{27}} = \frac{20\sqrt{54}}{27} = \frac{20*3\sqrt{6}}{27} = \frac{60\sqrt{6}}{27}$.
Simplifying further gives:
$= \frac{20\sqrt{6}}{9}$.
*****TASK CREATION AGENT PROMPT*****
You are to use the result from an execution agent to create new tasks with the following objective: Solve math problems. The last completed task has the result: First, let's simplify each square root separately:
- $$\sqrt{160} = \sqrt{16 * 10} = \sqrt{16} * \sqrt{10} = 4 * \sqrt{10}$$
**...Repeat after "First, let's simplify each square.."**
Now, substitute these values into the original expression which gives:
$$\frac{4 * \sqrt{10}}{2\sqrt{63}} \times \frac{7\sqrt{5}}{6\sqrt{3}} = \frac{2 * \sqrt{10}}{\sqrt{63}} \times \frac{7\sqrt{5}}{2\sqrt{3}}$$
We get rid of the square root in the denominator by multiplying the fractions which gives:
$$= \frac{2 * 7\sqrt{10 * 5}}{2 * \sqrt{63 * 3}} = \frac{14\sqrt{50}}{\sqrt{189}} = \frac{14 * 5\sqrt{2}}{7\sqrt{27}} = 2\sqrt{2} * \frac{10}{\sqrt{27}} = \frac{20\sqrt{2}}{\sqrt{27}}.$$
To simplify further and rationalize the denominator, we multiply by $\frac{\sqrt{27}}{\sqrt{27}}$ which gives:
$$= \frac{20\sqrt{2}}{\sqrt{27}} * \frac{\sqrt{27}}{\sqrt{27}} = \frac{20\sqrt{54}}{27} = \frac{20 * 3\sqrt{6}}{27} = \frac{60\sqrt{6}}{27}.$$
Simplifying further gives:
$= \frac{20\sqrt{6}}{9}$. This result was based on this task description: Completely simplify and rationalize the denominator: $\frac{\sqrt{160}}{\sqrt{252}} \times \frac{\sqrt{245}}{\sqrt{108}}$. Based on the result, return a list of tasks to be completed in order to meet the objective. Return one task per line in your response. The result must be a numbered list in the format:
# First task # Second task
The number of each entry must be followed by a period. If your list is empty, write "There are no tasks to add at this time." Unless your list is empty, do not include any headers before your numbered list or follow your numbered list with any other output.
****TASK CREATION AGENT RESPONSE****
1. Simplify and rationalize the denominator for the expression $\frac{\sqrt{225}}{\sqrt{45}} \times \frac{\sqrt{200}}{\sqrt{125}}$ 2. Simplify and rationalize the denominator for the expression $\frac{\sqrt{289}}{\sqrt{361}} \times \frac{\sqrt{100}}{\sqrt{72}}$ **...Until 10**
Adding new tasks to task_storage 'task_name': 'Simplify and rationalize the denominator for the expression fracsqrt225sqrt45timesfracsqrt200sqrt125', 'task_id': 2
'task_name': 'Simplify and rationalize the denominator for the expression fracsqrt289sqrt361timesfracsqrt100sqrt72', 'task_id': 3
**...Until task id 11**
****TASK PRIORITIZATION AGENT PROMPT****
You are tasked with prioritizing the following tasks: Simplify and rationalize the denominator for the expression fracsqrt225sqrt45timesfracsqrt200sqrt125 Simplify and rationalize the denominator for the expression fracsqrt289sqrt361timesfracsqrt100sqrt72
**...Aborted** |

Table 16: Application A1, Scenario 1: Autonomous math problem-solving. Preliminary test with CAMEL. We constructed two roles: "Math Solver" and "Python Programmer". The resulting code cannot be executed. The conversation continues, and the roles are not keen on solving the problem and giving an answer.

| | Action | CAMEL |
|---|---|---|
| 👤 | Set task prompt | Completely simplify and rationalize the denominator:

$$\frac{\sqrt{160}}{\sqrt{252}} \times \frac{\sqrt{245}}{\sqrt{108}}$$ |
| 🤖 | System | **************Assistant sys message************** AI SystemMessage(role_name='Python Programmer', role_type=<RoleType.ASSISTANT: 'assistant'>, meta_dict='task': 'Using Python programming language, write a program that accepts four integers as input, applies square root function to each, simplifies and rationalizes the denominator of the fraction resulting from the multiplication of two fractions: $\frac{\sqrt{160}}{\sqrt{252}} \times \frac{\sqrt{245}}{\sqrt{108}}$, and returns the simplified expression.', 'assistant_role': 'Python Programmer', 'user_role': 'Math Solver', role='system', content='Never forget you are a Python Programmer and I am a Math Solver. Never flip roles! ...)
**************AI User sys message**************
SystemMessage(role_name='Math Solver', role_type=<RoleType.USER: 'user'>, meta_dict='task': 'Using Python programming language, write a program that accepts four integers as input, applies square root function to each, simplifies and rationalizes the denominator of the fraction resulting from the multiplication of two fractions: $\frac{\sqrt{160}}{\sqrt{252}} \times \frac{\sqrt{245}}{\sqrt{108}}$, and returns the simplified expression.', 'assistant_role': 'Python Programmer', 'user_role': 'Math Solver', role='system', content='Never forget you are a Math Solver and I am a Python Programmer. Never flip roles! You will always instruct me.We share a common interest in collaborating to successfully complete a task.I must help you to complete the task.Here is the task: Using Python programming language, write a program that accepts four integers as input, applies square root function to each, simplifies and rationalizes the denominator of the fraction resulting from the multiplication of two fractions: $\frac{\sqrt{160}}{\sqrt{252}} \times \frac{\sqrt{245}}{\sqrt{108}}$, and returns the simplified expression.. ...)
**************Original task prompt**************
Completely simplify and rationalize the denominator: $\frac{\sqrt{160}}{\sqrt{252}} \times \frac{\sqrt{245}}{\sqrt{108}}$
**************Specified task prompt**************
Using Python programming language, write a program that accepts four integers as input, applies square root function to each, simplifies and rationalizes the denominator of the fraction resulting from the multiplication of two fractions: $\frac{\sqrt{160}}{\sqrt{252}} \times \frac{\sqrt{245}}{\sqrt{108}}$, and returns the simplified expression.
**************Final task prompt**************
Using Python programming language, write a program that accepts ... |
| 👤 | AI User | Instruction: Import the necessary libraries for the task. Input: None |
| 🤖 | AI Assistant | Solution: To solve this task, we will need the 'math' library for the square root function and the 'fractions' library to simplify the resulting fraction. Here is how to import these libraries:

```python\nimport math\nfrom fractions import Fraction\n```

The 'math' library provides functions that deal with mathematical operations. The 'sqrt' function, which we will use in this task, is part of this library.
The 'fractions' library provides functions for creating and manipulating rational numbers. The 'Fraction' function, which we will use to simplify the resulting fraction, is part of this library.
Next request. |
| 👤 | AI User | Instruction: Define a function that accepts four integers as input. Input: None |
| 🤖 | AI Assistant | Solution: Let's define a function named 'simplify_fraction' that accepts four integers as input. These integers will represent the numbers under the square root in the fractions.

```python\ndef simplify_fraction(a, b, c, d):\n    pass\n```

In this function, 'a', 'b', 'c', and 'd' are the parameters that will accept the four integers as input. The 'pass' statement is a placeholder and will be replaced with actual code in the next steps.
Next request. |
| 👤 | AI User | Instruction: Inside the function, apply the square root function to each of the input integers. Input: None |
| 🤖 | AI Assistant | Solution: We will use the 'math.sqrt' function from the 'math' library to apply the square root to each of the input integers. Here is how to do it:

```python\ndef simplify_fraction(a, b, c, d):\n    sqrt_a = math.sqrt(a)\n    sqrt_b = math.sqrt(b)\n    sqrt_c = math.sqrt(c)\n    sqrt_d = math.sqrt(d)\n```

In this code, 'sqrt_a', 'sqrt_b', 'sqrt_c', and 'sqrt_d' are variables that store the square root of the input integers 'a', 'b', 'c', and 'd' respectively.
Next request. |
| 👤 | Human | Abort |

Table 17: Application A1, Scenario 1: Autonomous math problem-solving. Preliminary evaluation on MetaGPT. It does not solve the math problem but gives a product solution.

| | **Action** | MetaGPT |
|---|---|---|
| 👤 | Prompt | Completely simplify and rationalize the denominator: $$\frac{\sqrt{160}}{\sqrt{252}} \times \frac{\sqrt{245}}{\sqrt{108}}$$ |
| 🤖 | | The boss has tasked me with creating a successful product that is efficient. ## Product Goals |

```
1  [
2      ``Create a product that meets the needs of the target
       market",
3      ``Ensure the product is efficient and easy to use",
4      ``Achieve a high level of customer satisfaction"
5  ]
```

## User Stories

```
1  [
2      ``As a user, I want a product that is easy to
       navigate so that I can find what I need quickly",
3      ``As a user, I want a product that is reliable and
       efficient so that I can depend on it",
4      ``As a user, I want a product that meets my needs so
       that I can get value for my money",
5      ``As a user, I want a product that is constantly
       updated and improved so that I can enjoy the best
       features",
6      ``As a user, I want a product that offers good
       customer support so that I can get help when I need it
       "
7  ]
```

## Competitive Analysis

```
1  [
2      ``Product A: Offers similar features but lacks
       efficiency",
3      ``Product B: Highly efficient but does not meet all
       the needs of the target market",
4      ``Product C: Has a high level of customer
       satisfaction but is not easy to navigate",
5      ``Product D: Is constantly updated and improved but
       lacks reliability",
6      ``Product E: Offers good customer support but is not
       efficient",
7      ``Product F: Is reliable and efficient but does not
       offer good customer support",
8      ``Our Target Product: Aims to combine all the best
       features of the competitors while improving on their
       weaknesses"
9  ]
```

## Competitive Quadrant Chart

| 👤 | Human | Abort |

Table 18: Application A7. Example output of MiniWobChat on one web manipulation task "click-button-sequence."

| | Action | AutoGen |
|---|---|---|
| 👤 | Init chat | Executor agent to AssistantAgent: |

Executor agent to AssistantAgent:
We have an autonomous computer control agent that can perform atomic instructions specified by natural language to control computers. There are two types of instructions it can execute.
First, given the instruction that matches the regular expression ```^type\s.{1,}$", it can type a list of characters via the keyboard. This instruction should specify the target keyboard input for the agent to type. Before this typing instruction, you should first locate the cursor by clicking the input box with click instruction.
Second, given the instruction that matches the regular expression, ```^clickxpath\s.{1,}$", it can click an HTML element with an xpath that is visible on the webpage. The target of this instruction should be a valid xpath. Below is the HTML code of the webpage where the agent should solve a task.

```
1  <div id="wrap" data-wob_ref="2" data-wob_eps="e0">
2    <div id="query">Click button ONE, then click button TWO.</div>
3    <div id="area" data-wob_ref="3" data-wob_eps="e0">
4      <button id="subbtn" style="position:absolute; left:103px; top:87px" data-wob_ref
         ="4" data-wob_eps="e0">ONE</button>
5      <button id="subbtn2" style="position:absolute; left:44px; top:97px" data-wob_ref
         ="5" data-wob_eps="e0">TWO</button>
6    </div>
7  </div>
```

task: Click button ONE, then click button TWO.
plan:
1. Click the button with xpath "//button[@id='subbtn']".
2. Click the button with xpath "//button[@id='subbtn2']".
Current task: Click button ONE, then click button TWO.
plan:

---

🤖

************************************************************
AssistantAgent to Executor agent:
1. Click the button with xpath "//button[@id='subbtn']".
2. Click the button with xpath "//button[@id='subbtn2']".
************************************************************
Executor agent to AssistantAgent:
Below is the HTML code of the webpage where the agent should solve a task.

```
1  <div id="wrap" data-wob_ref="2" data-wob_eps="e0">
2    <div id="query">Click button ONE, then click button TWO.</div>
3    <div id="area" data-wob_ref="3" data-wob_eps="e0">
4      <button id="subbtn" style="position:absolute; left:103px; top:87px" data-wob_ref
         ="4" data-wob_eps="e0">ONE</button>
5      <button id="subbtn2" style="position:absolute; left:44px; top:97px" data-wob_ref
         ="5" data-wob_eps="e0">TWO</button>
6    </div>
7  </div>
```

Current task: Click button ONE, then click button TWO.
Here is a plan you are following now.
1. Click the button with xpath "//button[@id='subbtn']".
2. Click the button with xpath "//button[@id='subbtn2']".
We have a history of instructions that have been already executed by the autonomous agent so far.
No instruction has been executed yet.
Based on the plan and the history of instructions executed so far, the first instruction should be '
************************************************************
AssistantAgent to Executor agent:
clickxpath //button[@id='subbtn']
************************************************************
Executor agent to AssistantAgent:
Hold on, please wait for my next instruction.
************************************************************
AssistantAgent to Executor agent:
Sure, I'll be here when you're ready with your next instruction.
************************************************************
Executor agent to AssistantAgent:
Below is the HTML code of the webpage where the agent should solve a task.

```
1  <div id="wrap" data-wob_ref="2" data-wob_eps="e0">
2    <div id="query">Click button ONE, then click button TWO.</div>
3    <div id="area" data-wob_ref="3" data-wob_eps="e0">
4      <button id="subbtn" style="position:absolute; left:103px; top:87px" data-wob_ref
         ="4" data-wob_eps="e0" data-tampered="e0">ONE</button>
5      <button id="subbtn2" style="position:absolute; left:44px; top:97px" data-wob_ref
         ="5" data-wob_eps="e0">TWO</button>
6    </div>
7  </div>
```

Current task: Click button ONE, then click button TWO.
Here is a plan you are following now.
1. Click the button with xpath "//button[@id='subbtn']".
2. Click the button with xpath "//button[@id='subbtn2']".
We have a history of instructions that have been already executed by the autonomous agent so far.
1: clickxpath //button[@id='subbtn']
Based on the plan and the history of instructions executed so far, the next proper instruction should be '
************************************************************
AssistantAgent to Executor agent:
clickxpath //button[@id='subbtn2']
************************************************************
Executor agent to AssistantAgent:
SUCCESS!!!!

---

| 🤖 | | | TERMINATE |

