# OpenReview forum: "AutoGen: Enabling Next-Gen LLM Applications via Multi-Agent Conversations"
_colmweb.org/COLM/2024/Conference — COLM_

### Official Review · Reviewer_9adi · 2024-05-08

**Rating:** 8
**Confidence:** 5
**Ethics Flag:** 1

**Summary:**

The paper introduces AutoGen, an open-source framework that enables multiple LLM-based agents to collaborate on tasks through role-playing and tool usage. The framework has gained significant attention from the community since its release. While the central idea of multi-agent interaction in LLMs aligns with prior work and some earliy frameworks like SocraticAI [1], the authors highlight two novel engineering concepts: "conversable agents" and "conversation programming," which simplify agent configuration and interaction for developers. AutoGen also unifies other multi-agent LLM frameworks, and the paper demonstrates its wide usability through six application examples, along with quantitative evaluations and comparisons to other methods.

The engineering effort behind AutoGen is commendable, and this open-source framework is a valuable resource for the COLM community. However, the authors should temper their claims about the absolute superiority of multi-agent methods for problem-solving and discuss the limitations of these methods, such as potential failure modes and comparisons of inference costs. Additionally, the paper would benefit from a more rigorous validation of the effectiveness and key design factors of multi-agent systems, such as the separation of context among agents and the use of directed graphs versus broadcasting.

There are some controversy surrounding the superiority of multi-agent systems over single agents [2], as well as the potential for agents to be distracted by irrelevant context [3]. While a comprehensive validation of multi-agent superiority may be beyond the scope of this paper, AutoGen provides an efficient platform for conducting comparative experiments to test the effectiveness and key design factors of multi-agent task-solving systems. The authors should provide more caveats to readers regarding these aspects.

[1] SocraticAI: https://princeton-nlp.github.io/SocraticAI/
[2] Huang & Chen et al. Large Language Models Cannot Self-Correct Reasoning Yet. ICLR'24
[3] Shi & Chen et al. Large Language Models Can Be Easily Distracted by Irrelevant Context ICML'23

**Reasons To Accept:**

- novel engineering design of the framework
 - thorough experiments demonstrating wide application
 - open-source, helpful resource to the community

**Reasons To Reject:**

- quantitative evaluation can be improved to show statistical significance of the results. figure 4, in particular, should include sample sizes and appropriate error measurements.
- the effectiveness of multi-agent over single agent should be more carefully examined. the performance gap in experiment A1 is mainly from tool usage, and it's unclear whether it's human-in-the-loop scenario.
 - didn't do enough analyses of failure modes in A1-A6.

---

> ### Author Rebuttal · Authors · 2024-05-31
>
> We thank the reviewer for the constructive feedback and thoughtful questions. We also appreciate your recognition of the novel engineering design and extensive application scope of AutoGen. Here, we provide further clarifications and additional data in response to the weaknesses highlighted.
>
> ### **1. About sample sizes and significance of quantitative evaluation:**
> The results presented in Figure 4 are derived from widely recognized datasets, ensuring the relevance and robustness of our findings: MATH (used in A1) includes 5,000 test instances; Natural Questions (in A2) includes 7000+ test instances; ALFWorld (in A3) includes 134 unseen test instances; Supply-chain related Coding tasks [4] (used in A4) includes 100 data instances.
>
> ### **2. About the effectiveness of multi-agent design:**
> In addition to A1, we also investigate the effectiveness of the multi-agent approach in A4, where a multi-agent approach is compared with a single agent conducting both the code-writing and safeguard processes. The performance improvement is shown in Figure 4(d). As for the human-in-the-loop scenario, we consider it a pilot study and defer comprehensive evaluation to future work, as the evaluation involves a human study.
> ### **3. About failure analysis:**
> Failure analyses are included in the appendix to provide insights into the limitations and improvement areas for AutoGen. These analyses are presented across various contexts: Table 1 (failure analysis of AutoGen and compared methods on MATH), Figure 8 (failure analysis of RAGChat with and without interactive retrieval), Figure 10 (failure analysis of a two-agent system compared to a three-agent system in ALFWorld), Figure 13 (failure analysis of a two-agent chat compared to a group chat), Figure 16 (failure analysis of conversational chess when a board agent is not used), and Table 6 (failure analysis of AutoGen on 4 typical tasks from MiniWob+)
>
> We thank the reviewer again for the time and effort. We hope this additional information adequately addresses your concerns. We are very willing to address any further questions or concerns you might have.

---

> > ### Comment · Reviewer_9adi · 2024-06-04
> >
> > Would be better to report error bars in Figure 4. How many runs for each instance? How do you set temperatures?

---

> > ### Author Response · Authors · 2024-06-06
> > **Details about figure 4**
> >
> > Thank you for your thoughtful comment! We are definitely mindful of the significance of the empirical results.
> >
> > For results about A1 and A2 reported in Figure 4, we are doing a single run on each instance. Considering the large number of test instances (5000 and 7000), making multiple runs is in fact unnecessary and not that practical (note that running all the methods with GPT-4 costs several thousand dollars per run). To make this point more concrete, we are conducting this evaluation according to the standard procedure established in the original papers about these two datasets [1][2], and following the practices from recent well-received papers with state-of-the-art baselines [3]. Specifically in A2, we are directly comparing the results from Adlakha et al. [3], which include results from only one run. We are using the default temperature in the OpenAI API.
> >
> > For A3, we are following the exact same experimental setting as the ReAct [4] paper and include **3 runs for each method**. The temperature is set to 0.
> >
> > For A4, considering the number of data instances is not that large, we add more runs, and the aggregated results (including both average and error bars) from 10 total runs are reported below:
> >
> > | Model        | Multi/Single-Agent | Accuracy (mean ± std) | F1 Score (mean ± std) |
> > |--------------|--------------|-----------------------|-----------------------|
> > | GPT-3.5-Turbo| Multi           | 0.786 ± 0.011         | 0.829 ± 0.008         |
> > | GPT-3.5-Turbo| Single          | 0.707 ± 0.019         | 0.596 ± 0.038         |
> > | GPT-4        | Multi           | 0.901 ± 0.012         | 0.906 ± 0.011         |
> > | GPT-4        | Single          | 0.879 ± 0.012         | 0.862 ± 0.015         |
> >
> > Please feel free let us know if there are more questions. We are happy to address any further questions you might have! We really appreciate your constructive comments!
> >
> > [1] Hendrycks, Dan, Collin Burns, Saurav Kadavath, Akul Arora, Steven Basart, Eric Tang, Dawn Song, and Jacob Steinhardt. "Measuring mathematical problem solving with the math dataset." arXiv preprint arXiv:2103.03874 (2021).
> >
> > [2] Kwiatkowski, Tom, et al. "Natural questions: a benchmark for question answering research." Transactions of the Association for Computational Linguistics 7 (2019): 453-466.
> >
> > [3] Adlakha, Vaibhav, Parishad BehnamGhader, Xing Han Lu, Nicholas Meade, and Siva Reddy. "Evaluating correctness and faithfulness of instruction-following models for question answering." arXiv preprint arXiv:2307.16877 (2023).
> >
> > [4] Yao, Shunyu, Jeffrey Zhao, Dian Yu, Nan Du, Izhak Shafran, Karthik Narasimhan, and Yuan Cao. "React: Synergizing reasoning and acting in language models." arXiv preprint arXiv:2210.03629 (2022).

---

> > > ### Comment · Reviewer_9adi · 2024-06-06
> > >
> > > Thanks! The experiment setup makes sense. I’ve increased my score.

---

### Official Review · Reviewer_TpLJ · 2024-05-10

**Rating:** 3
**Confidence:** 4
**Ethics Flag:** 1

**Summary:**

This paper presents an open-source framework to solve tasks by combining multiple LLMs acting as agents that talk to each other. It describes necessary features of such a framework, such as being easy to customize (to allow generality) or allowing re-use of modules (to reduce complexity). The main proposal is the concept of "conversation programming" which implies the definition of a set of agents and the definition the code that controls their interaction. After the main proposal the paper describes the features of the framework and the applications, which contains various experimental setups.

**Reasons To Accept:**

The topic is timely: agents and the authors are straightforward about the paper is being an open source framework.

**Reasons To Reject:**

My main problem is that it is hard to judge the contributions of this work from a research perspective both because, as the abstract states, the papers main proposal is an open source framework and also because the experimental setup needs a lot of improvement. The proposal of "conversation programming" seems also vague and its not clear how it is different from a generic multi-agent framework.

To put it in concrete terms:

- Most of the manuscript reads more like the documentation of a software package than a research paper. It contains a lot of description of software abstractions such as classes and general descriptions of the package's features and even usage of flags or variable values. The software-like description makes it hard to isolate the concrete components that contribute to the experimental setup's results.

- The experimental setup itself is also lacking. Many of the results are also missing from the main paper and moved to the Appendix e.g RaG. These results are important enough that they should be on the main body of the manuscript. The experiments in the Appendix seem also not standard, using a small number examples and oddly organized with a general lack of tables.

- It seems like the original manuscript was far longer ans it has been split between body and appendix in a way that makes it hard to follow.

---

> ### Author Rebuttal · Authors · 2024-05-31
>
> We thank the reviewer's time and effort in evaluating our submission. We appreciate your feedback and would like to address them!
>
> ### **1. About Open-Source Framework Contribution**:
> We assert that the development of an open-source framework constitutes a significant research contribution. Our framework introduces innovative engineering designs and serves as a vital resource to the community, as acknowledged by all the other three reviewers. It facilitates scientific experimentation, yielding useful insights and enabling new discoveries. This is also evidenced by the wide adoption of the framework and the numerous research works developed based on it. A notable example of a similar type of open-source framework is PyTorch [1]. PyTorch has been cited over 42,405 times since its publication at NeurIPS 2019, underscoring its profound impact across various research domains. Regarding the “ software-like description”: The PyTorch paper also includes detailed descriptions of software architecture (refer to Listings 1 and 2). These kinds of descriptions are generally helpful in outlining the framework's engineering design and architecture.
>
> ### **2. About Experimental Setup and Results in the Appendix, and Organization of the Paper**
> Due to page limitations, detailed experimental setups and extensive results were placed in the appendix. Nevertheless, the main body of the paper encapsulates all essential details needed to grasp the experimental framework, supported by robust results and illustrative figures in Section 3. The results highlighted in the appendix are meant for readers interested in a deeper exploration, ensuring the main text remains focused and readable. These additional details and results are indeed appreciated by most readers, including all the other three reviewers. With that being said, we do appreciate the feedback on improving our paper's organization and are committed to enhancing its clarity to better convey our findings.
>
> [1] Paszke, Adam, et al. "Pytorch: An imperative style, high-performance deep learning library." Advances in neural information processing systems 32 (2019).
>
> We hope these clarifications address your concerns and are willing to address any more concerns you might have. We will further improve the paper based on your feedback.

---

> > ### Comment · Reviewer_TpLJ · 2024-06-05
> >
> > I thank the authors for their detailed answer

---

> > > ### Author Response · Authors · 2024-06-06
> > > **Thank you!**
> > >
> > > Thank you for your review of our paper. We greatly appreciate the time and effort you have invested in evaluating our work. Should you have any further questions or need additional clarifications, we are more than willing to provide them. We hope that our responses have addressed your concerns satisfactorily. We hope you consider raising the rating on our paper. Thank you once again for your valuable input!

---

### Official Review · Reviewer_Z1Mk · 2024-05-11

**Rating:** 6
**Confidence:** 3
**Ethics Flag:** 1

**Summary:**

The paper proposes AutoGen, an open-source framework for constructing Large Language Models (LLMs), characterized by utilizing conversations among multiple agents to build specific applications, also referred to as conversational programming. In terms of evaluation, the paper covers six different use cases, including mathematical problem-solving and question-answering tasks, demonstrating the effectiveness of AutoGen. Overall, AutoGen is a promising framework for the rapid, flexible, and customizable development of LLM applications.

**Questions To Authors:**

None

**Reasons To Accept:**

1. It is a fully open-source multi-agent transformation framework, facilitating users in rapid application development. The high transparency of technical details aids in traceability and reproducibility.
2. The authors provide ample empirical results to showcase the performance of multi-agent conversations, aiding readers in better understanding the advantages and limitations of multi-agent systems.

**Reasons To Reject:**

1. The claimed multi-agent system is more accurately described as multi-prompting. More convincing evidence would be provided by observing dialogues under different conditions within the agents themselves.
2. The major concern lies in the inherent differences from other multi-agent frameworks, such as CAMEL (NeurIPS 2023) and MetaGPT (ICLR 2023). In the related work and discussion, the paper assert no explicit restrictions on the number of agents and whether tools are supported to distinguish from previous works. In my opinion, these are not very strong justifications.

---

> ### Author Rebuttal · Authors · 2024-05-31
>
> We sincerely thank the reviewer for the insightful comments and thoughtful questions.  We also truly appreciate your recognition of the open-source contribution, technical details, and rich empirical results!
>
> 1. A multi-agent system could possibly be considered to be performing multi-prompting when all the agents are LLM-backed. However, the AutoGen framework goes beyond multi-prompting: in addition to LLM, an agent could also be backed by tools or humans, or any combination of them. We provide example dialogues under different design choices in Figures 8, 10, 13, and 16.
>
> 2. As for the difference between other existing work: In addition to the restrictions on the number of agents, CAMEL and MetaGPT essentially only support a fixed multi-agent workflow in a round-robin fashion or a pre-defined SOP fashion (both are static conversation patterns). AutoGen supports a diverse workflow (static or dynamic) thanks to its composable conversation patterns. This is vital in enabling diverse application scenarios, where we may want to orchestrate agents statically or dynamically depending on application needs.
>
> We hope this addresses your concerns, and we are very willing to respond to any further questions you might have!

---

> > ### Comment · Reviewer_Z1Mk · 2024-06-06
> >
> > Thanks for response. I would like to maintain my current score.

---

### Official Review · Reviewer_sCwk · 2024-05-11

**Rating:** 6
**Confidence:** 4
**Ethics Flag:** 1

**Summary:**

This paper proposes an open-source framework that allows developers to build LLM applications by composing multiple agents to converse with each other to accomplish tasks. AutoGen provides a few default agents with preset behaviors and it also provides users the ability to create flexible agent behaviors and conversation patterns for different applications using both natural language and code. Autogen and their preset agents are tested in multiple applications covering various knowledge domains. Results show that autogen’s performance is always better  than single LLM methods.

**Reasons To Accept:**

1. Research-wise it serves as another piece of evidence suggesting that multi-agent collaboration might be an effective method to build successful LLM applications.

2. A solid open-source system that allows users to not only adopt pre-defined agents freely, but also to contribute more types of agents to potentially form an ecosystem.

3. Solid experiments covering multiple applications and various knowledge domains.

**Reasons To Reject:**

My only concern is I don’t know how easy it is for the multi-agent system to achieve the reported superior performance compared to single LLM solutions. Do you have to try a lot of different designs to make it happen or your first several intuitive designs would work? Are there difficulty level differences between domains for autogen? In other words, have you observed that for some applications, you really have to try very hard and try a lot of different designs to make it work better than GPT-4 while for other apps it was very easy? Analysis towards this direction can be very helpful.

---

> ### Author Rebuttal · Authors · 2024-05-31
>
> We thank the reviewer for the insightful comments and thoughtful questions. We truly appreciate your recognition of AutoGen's contribution as a robust open-source system and the extensive experiments covering multiple applications and various knowledge domains.
>
> Regarding your query about the difficulty of achieving superior performance with our multi-agent system compared to single LLM solutions: We observe that AutoGen’s built-in agents, such as the AssistantAgent equipped with coding capabilities, alongside a user proxy agent with a Python interpreter, typically perform well across a broad range of general domain tasks and mathematical problem-solving. As evidenced by the results shown in Figure 4(a), even a straightforward two-agent system can achieve superior outcomes without extensive customization. For specialized tasks requiring domain expertise, several iterations might be needed to optimize performance. However, our flexible "group chat" conversation pattern (as introduced in A5 of Section 3) significantly simplifies this process. This feature allows for the dynamic selection of agents based on real-time conversation history and task requirements, thereby enhancing the adaptability of our system to different complexities.
>
> We hope this addresses your question, and we are very willing to elaborate more if the clarification is unclear.

---

> > ### Author Response · Authors · 2024-06-06
> > **Thank you and some more data!**
> >
> > Thank you for your thoughtful review and for the insightful questions.
> >
> > To address your question further: We generally observe that multi-agent systems tend to achieve superior performance compared to single-agent solutions, except in tasks that are inherently simple. To elucidate this point, we conducted a survey within the community utilizing AutoGen, which garnered 565 responses. This survey provides a general sense of the domains where multi-agent designs are considered most beneficial.
> >
> > | Application Category          | Percentage of use (based on 565 responses)  |
> > |------------------------------|------------|
> > | Software Development         | 20.53%     |
> > | Agent Platform               | 9.91%      |
> > | Research                     | 9.03%      |
> > | Data Processing              | 7.79%      |
> > | Consulting                   | 6.37%      |
> > | Content Creation             | 6.37%      |
> > | Finance                      | 6.02%      |
> > | Web Browsing                 | 5.49%      |
> > | Healthcare                   | 5.13%      |
> > | Operating System Automation  | 4.60%      |
> > | Marketing                    | 4.42%      |
> > | Information Security         | 4.07%      |
> > | Education                    | 3.72%      |
> > | Legal                        | 2.48%      |
> > | Sales                        | 1.95%      |
> > | Gaming and Simulation        | 1.59%      |
> > | Mobile                       | 0.53%      |
> >
> > We hope that our responses have addressed your questions satisfactorily, and hope you consider raising the rating on our paper.  Thank you once again for your valuable input and for the opportunity to clarify these points.

---

> ### Author Response · Authors · 2024-06-07
>
> Dear Reviewer sCwk,
>
> Thank you again for reviewing our work, and for acknowledging the open-source contribution and the solidity of our experimental study. As the discussion deadline approaches, we are writing to kindly ask if the questions raised in your original review have been addressed, especially with the additional data from the user survey added. If so, we hope you can raise the rating for our submission. We are very keen to address any other questions you might have.
>
> Thank you!

---

### Decision · Program_Chairs · 2024-07-10

**Decision:**

Accept

**Comment:**

This work introduces AutoGen, an open-source framework that allows developers to build LLM applications using multiple conversable agents and conversation programming.

As reviewers pointed out, this work presents a strong open-source framework, and their extensive experimentation demonstrates the utility of the proposed framework.

Reviewers also pointed out a few concerns, such as how to optimize the multi-agent systems to let it work better than single LLMs, the differences of AutoGen to other multi-agent frameworks, as well as statistical significance and error analyses.  We thank the authors for their detailed response during the rebuttal period, and some of these concerns were addressed.

Another issue got discussed was the contribution of this work, as one reviewer questioned that the current work reads more like a documentation of software package. I think open-source framework is indeed a contribution, and the resulting open-source work can be a useful resource for the community.

There are also some other issues such as the framing of conversation prompting, for which we encourage the authors to further address these in the revised version.